


# Towards Strongly-coupled Ensemble Data Assimilation with Additional Improvements from Machine Learning

Eugenia Kalnay[1, 2], Travis Sluka[3], Takuma Yoshida[4], Cheng Da[5, 6], and Safa Mote[1, 2]

5  [1] Department of Atmospheric and Oceanic Science, University of Maryland, College Park, College Park, Maryland

[2] Institute for Physical Science and Technology, University of Maryland, College Park, College Park, Maryland

[3] Joint Center for Satellite Data Assimilation, University Corporation for Atmospheric Research, Boulder, Colorado

[4] Numerical Prediction Development Center, Japan Meteorological Agency, Japan

[5] Earth System Science Interdisciplinary Center, University of Maryland, College Park, College Park, Maryland

10  [6] NASA Global Modeling and Assimilation Office (GMAO), Greenbelt, Maryland

*Correspondence to*: Eugenia Kalnay (ekalnay@umd.edu), Cheng Da (cda@umd.edu), Safa Mote (ssm@umd.edu)

**Abstract.** We assessed different coupled data assimilation strategies with a hierarchy of coupled models, ranging from the simple coupled Lorenz model to the state-of-the-art coupled general circulation model CFSv2. With the coupled Lorenz model, we assessed the analysis accuracy by strongly-coupled Ensemble Kalman Filter (EnKF) and 4D-Variational (4D-Var) methods with varying assimilation window lengths. The analysis accuracy of the strongly-coupled EnKF with a short assimilation window is comparable to that of 4D-Var with a long assimilation window. For 4D-Var, the strongly-coupled approach with the coupled model produces more accurate ocean analysis than the ECCO-like approach using the uncoupled ocean model. Experiments with the coupled quasi-geostrophic model conclude that the strongly-coupled approach outperforms the weakly-coupled and uncoupled approaches for both the full-rank EnKF and 4D-Var, with the strongly-coupled EnKF and 4D-Var showing a similar level of accuracy higher than other coupled data assimilation approaches such as the outer loop coupling. A strongly-coupled EnKF software framework is developed and applied to the intermediate-complexity coupled model SPEEDY-NEMO and the state-of-the-art operational coupled model CFSv2. Experiments assimilating synthetic or real atmospheric observations into the ocean through strongly-coupled EnKF show that the strongly-coupled approach improves the analysis of the atmosphere and upper oceans, but degrades observation fits in the deep ocean, probably due to the unreliable error correlation estimated by a small ensemble. The correlation-cutoff method is developed to reduce the unreliable error correlations between physically irrelevant model states and observations. Experiments with the coupled Lorenz model demonstrate that strongly-coupled EnKF informed by the correlation-cutoff method produces more accurate coupled analyses than the weakly-coupled and plain strongly-coupled EnKF regardless of the ensemble size. To extend the correlation-cutoff method to operational coupled models, a neural network approach is proposed to systematically acquire the observation localization functions for all pairs between the model state and



observation types. The following strongly-coupled EnKF experiments with an intermediate-complexity coupled model show promising results with this method.

## 1 Introduction

Coupled data assimilation (CDA) has drawn tremendous attention recently among the weather and climate modeling community [Penny et al., 2017]. It has been recognized as one of the most active research areas for data assimilation from now to the future [Carrassi et al., 2018]. Among the many benefits of exploring CDA [Penny et al., 2017, Penny and Hamill, 2017; Zhang et al., 2020], one primary motivation is the need to initialize the coupled models with the coupled analyses. Many operational centers have plans to make seamless weather–climate prediction using coupled general circulation models

[CGCM, Palmer et al., 2008; Hoskins, 2013], of which initialization requires analyses of different earth components (e.g., atmosphere, ocean, land, and ice). Past studies [Mulholland et al., 2015] show that the uncoupled data assimilation (UCDA) approach, which obtains independent analyses of different earth components based on the forecasts from uncoupled models, fails to produce balanced and physically consistent coupled analyses. The forecasts initialized from these uncoupled analyses suffer from severe initialization shocks. Zhang et al. [2007] adopted the weakly coupled data assimilation (WCDA) approach

by creating separate analyses of the atmosphere and oceans, assimilating their domain observations based on the forecasts initialized from a coupled model. They found that the WCDA approach could produce balanced coupled analyses that correctly reconstruct the variability and trends of the ocean in the 20th century. Through experiments with an intermediate-complexity atmosphere–ocean coupled model and a state-of-the-art coupled model, Sluka et al. [2016; 2018] found that the strongly-coupled data assimilation (SCDA) approach, which creates coupled analyses by assimilating the same set of the all-

domain observations into different earth components, outperforms the WCDA approach in terms of the analysis accuracy and observation departures.

Given the benefits of CDA, most operational centers are transitioning from UCDA to CDA [Penny and Hamill, 2017]. The National Center for Environmental Prediction (NCEP) pioneered producing the coupled analyses using a WCDA system that integrates the CGCM Climate Forecast System [CFS, Saha et al., 2006, 2010] and generates separate 3D-Var

analyses for the atmosphere and oceans. Suguira et al. [2008] implemented the full adjoint of a coupled general circulation model and used it to develop a 4D-Var SCDA system, with the initial ocean states and the bulk adjustment factors of surface fluxes as its analyzed variables. This approach is superior to the WCDA approach since it can directly update the coupled states with cross-domain observations through the backward integration of the adjoint for the fully-coupled model. However, this approach has not been widely adopted due to the technical challenge of developing and maintaining the adjoint of a

CGCM. Instead, most operational centers producing variational analyses adopted the WCDA approach, allowing them to reuse their existing separate atmosphere and ocean analysis systems [Lea et al., 2015; Browne et al., 2019]. The European Centre for Medium-Range Weather Forecasts (ECMWF) implemented a Quasi-SCDA system through the "outer loop





coupling", where the incremental 4D-Var atmospheric and 3D-FGAT oceanic analyses share the same outer loops so that their updated analyses will be used together to acquire the new model trajectory for the next round [Laloyaux et al., 2016; 2018]. Fujii et al. [2020] recently developed a Quasi-SCDA system MRI-CDA1 which applied different assimilation window lengths to produce atmospheric and oceanic analyses. Besides model development activities of variational CDA systems at operational centers, Smith et al. [2015, 2017, 2018, 2020] comprehensively examined the advantages of SC 4D-Var over other variational CDA approaches by using a single-column coupled model.

For the EnKF-based CDA systems for complex coupled models, Zhang et al. [2005, 2007] pioneered the development of an online EnKF-based CDA system for the Geophysical Fluid Dynamics Laboratory (GFDL) second-generation Coupled Model (CM2), and demonstrated that this WC EnKF could reconstruct the variability and trends of the ocean correctly in the 20th century. Lu et al. [2015a; 2015b] proposed to assimilate the lagged averaged high-frequency atmospheric observations into the ocean to increase the signal-to-noise ratio for the coupled analyses. They proved the effectiveness of this method for improved coupled analyses with an intermediate-complexity CGCM. Sluka et al. [2016] implemented offline WC and SC Local Ensemble Transform Kalman Filters (LETKFs) for an intermediate-complexity atmosphere–ocean coupled model and conducted identical twin experiments by assimilating synthetic atmospheric observations into the ocean through SC LETKF. Their results show that SCDA with the LETKF produces more accurate ocean and atmosphere analyses than WCDA. Sluka [2018] developed a prototype offline CDA system CFSv2-LETKF for the state-of-the-art coupled model CFSv2 that can be configured in either the WCDA or SCDA mode. The actual observation experiments with 50-member CFSv2-LETKF showed that SCDA improves the observation fits for the lower atmosphere and upper ocean but degrades the fits in the deep ocean. Karspeck et al. [2018] implemented an offline WC Ensemble Adjustment Kalman Filter (EAKF) system for the Community Earth System Model (CESM) and used this system to create a 12-year coupled reanalyses from 1970 to mid 1982. Besides the efforts to develop the EnKF-based CDA systems for complex coupled models, many challenges related to CDA have been recognized using low-order coupled models, which are summarized by Penny et al. [2017] and Zhang et al. [2020].

This paper reviews our efforts in exploring the benefits of SCDA over other CDA strategies using a wide range of coupled models with increasing complexities. We identified one issue of SC EnKF that can significantly degrade SC EnKF analyses and proposed a solution. In Section 2, we start our discussion with a coupled Lorenz model [Peña and Kalnay, 2004], investigating the capability of SCDA to constrain the slow and fast modes of a coupled system for both ensemble and variational methods simultaneously. In Section 3, we contrast the performance of SC 4D-Var and ECCO-like 4D-Var for ocean analysis in the coupled Lorenz system. Section 4 compares the analysis accuracy of ensemble and variational CDA methods with different CDA strategies by using a coupled Quasi-Geostrophic model. In Sections 5 and 6, we focus on developing EnKF-based CDA systems for complex coupled models (i.e., SPEEDY-NEMO and CFSv2) and comparing the performance of SCDA and WCDA in producing coupled analyses. In Section 7, we review the correlation-cutoff method that significantly improves the SC EnKF analysis when using a small ensemble, and discuss the experimental results with the





coupled Lorenz model. Section 8 shows how to take advantage of neural networks to extend the correlation-off method to an intermediate-complexity CGCM. Section 9 gives the summary and discussion.

## 2. CDA Experiments with the coupled Lorenz model

In this section, we discuss results obtained by Singleton [2011] who evaluated the capability of 4D-Var and EnKF in producing coupled analyses with a multi-scale coupled Lorenz system [Peña and Kalnay, 2004]. Different approaches are proposed to enhance those two types of assimilation methods for CDA.

For the CDA experiments, Singleton [2011] adopted the 9-variable coupled Lorenz system developed by Peña and Kalany [2004], of which equations are written as.

$$\dot{x}_e = \sigma(y_e - x_e) - c_e(Sx_t + k_1) \tag{1}$$

$$\dot{y}_e = rx_e - y_e - x_e z_e + c_e(Sy_t + k_1) \tag{2}$$

$$\dot{z}_e = x_e y_e - bz_e \tag{3}$$

$$\dot{x}_t = \sigma(y_t - x_t) - c(SX + k_2) - c_e(Sx_e + k_1) \tag{4}$$

$$\dot{y}_t = rx_t - y_t - x_t z_t + c(SY + k_2) + c_e(Sy_e + k_1) \tag{5}$$

$$\dot{z}_t = x_t y_t - bz_t + c_z Z \tag{6}$$

$$\dot{X} = \tau\sigma(Y - X) - c(x_t + k_2) \tag{7}$$

$$\dot{Y} = \tau rX - \tau Y - \tau SXZ + c(y_t + k_2) \tag{8}$$

$$\dot{Z} = \tau SXY - \tau bZ - c_z z_t \tag{9}$$

where $[x_e, y_e, z_e]^T$, $[x_t, y_t, z_t]^T$, and $[X, Y, Z]^T$ are the state vectors of the extratropical atmosphere, tropical atmosphere, and
tropical ocean, respectively. For this system, the tropical atmosphere is strongly coupled with the tropical ocean ($c=c_z=1$) but weakly coupled with the extratropical atmosphere ($c_e=0.08$). Meanwhile, no direct coupling occurs between the extratropical atmosphere and the tropical ocean. Other parameters of this model are $(\sigma, r, b, \tau, S, k_1, k_2) = (10, 28, \frac{8}{3}, 0.1, 1, 10, -11)$. Though simple, this coupled Lorenz system presents multi-scale dynamics and can reproduce "ENSO-like" oscillations for its tropical atmosphere and ocean, making it an ideal testbed for studying predictability and developing data assimilation
strategies for CDA [Peña and Kalnay, 2004; Norwood et al., 2013; Norwood et al. 2015; Yoshida and Kalnay, 2018; Yoshida 2019]. Singleton [2011] obtained the nature run by integrating the model using the 4th-order Runge-Kutta method with a time step $\Delta t = 0.01$. Observations are generated every 8 time-steps by adding to the true model states the uncorrelated Gaussian errors with zero mean and a standard deviation of $\sqrt{2}$. Besides, Assimilation experiments with the Ensemble Transform Kalman Filter (ETKF) in this section uses 9 members.

Singleton [2011] found that SC ETKF has the smallest analysis Root Mean Square Error (RMSE) when adopting an assimilation interval of 8 time-steps. Using longer assimilation intervals for the SC ETKF degrades the coupled analyses and causes the filter divergence eventually, consistent with the finding by Kalnay et al. [2007] that the EnKF prefers short assimilation intervals. Adopting 4D-ETKF [Hunt et al., 2004] or Quasi-Outer Loop [ETKF-QOL, Yang et al., 2012] allows





the SC ETKF to utilize long assimilation intervals and improve the coupled analyses (Figure 1). Separate ETKF analyses for
the fast (i.e., extratropical and tropical atmosphere) and slow modes (e.g., tropical ocean, corresponding to the "Atmospheric
coupling" pattern in Yoshida and Kalnay [2018]) show lower analysis error than the SC ETKF, especially when adopting
longer assimilation intervals. Among all ETKF-based methods, SC ETKF-QOL using a short assimilation interval of 8 time-
steps gives the most accurate analysis.

Figure 1 also presents the analysis errors for SC 4D-Var that adopts varying assimilation window lengths. Unlike
ETKF, SC 4D-Var analyses with longer assimilation window length generally show lower analysis errors, consistent with
the findings by Kalnay et al. [2007]. However, the optimal assimilation window lengths for different Lorenz subsystems are
different: the 4D-Var analysis error for the extratropical atmosphere starts to increase if the assimilation window length
exceeds 72 time-steps. Singleton [2011] found that such degradation caused by long assimilation window length is due to the
multiple minima during the minimization procedure. Implementing the Quasi-Static Variational Data Assimilation [QVA,
Pires et al., 1996; Kalnay et al., 2007] to SC 4D-Var avoid such degradation and allows the 4D-Var to utilize an even longer
assimilation window to improve the coupled analyses.

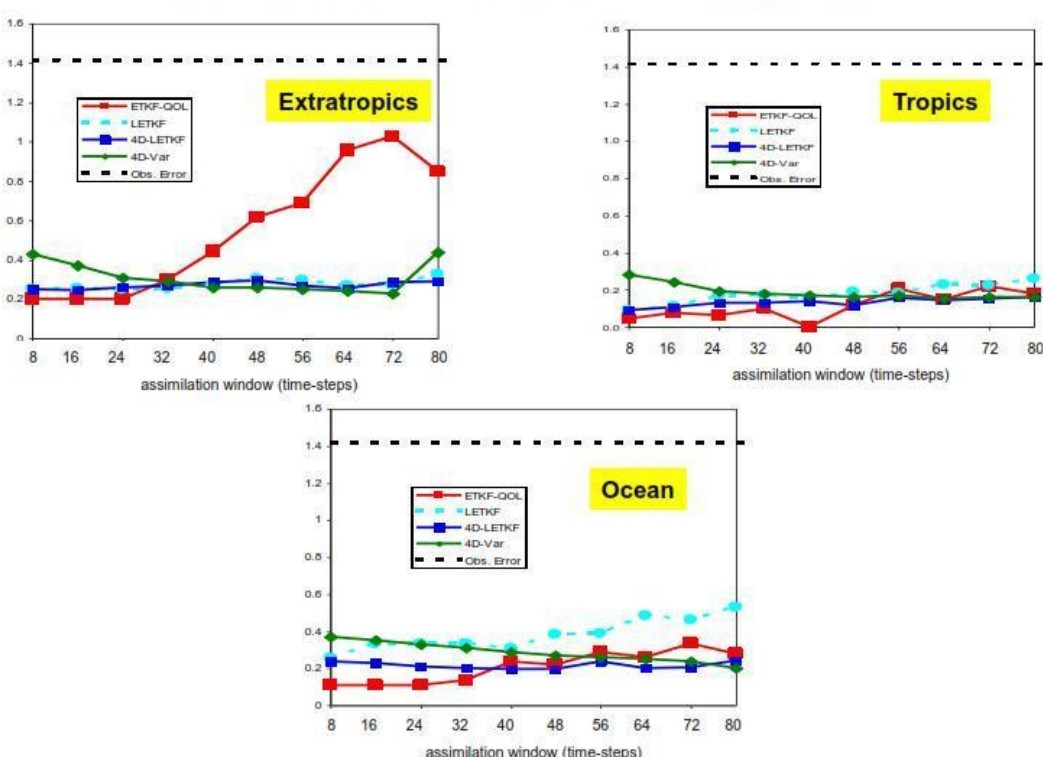

**Figure 1:** Time-averaged Analysis RMSE for SC 4D-Var (green), SC ETKF-QOL (red), SC ETKF with the "atmos-
145         coupling" (Figure 2 of Yoshida and Kalnay, 2018) as the localization pattern (cyan) and its 4D extension (blue) for





the extratropical atmosphere (top left), tropical atmosphere (top right), and ocean (bottom). Adapted from Singleton [2011]. Note that "ETKF" and "LETKF" in this figure (Singleton [2011]) refers to the SC ETKF, and SC ETKF with the "atmos-coupling" as the localization pattern based on the definition of CDA by Penny et al. [2017].


## 3. Comparison of the SC and the ECCO-like 4D-Var

Unlike ordinary 4D-Var that uses the initial model states as the analyzed variables, the ocean analysis Estimating the Circulation and Climate of the Ocean [ECCO; Stammer et al., 2004; Forget et al. 2015; Fukumori et al. 2017] includes additional surface forcing fields and mixing parameters as the analyzed variables in the 4D-Var cost function (Figure 2). The

approach allows ECCO to use an extremely long assimilation window of 10 years [Stammer et al., 2004], during which the ocean analysis is guaranteed to conserve momentum, heat and salinity.

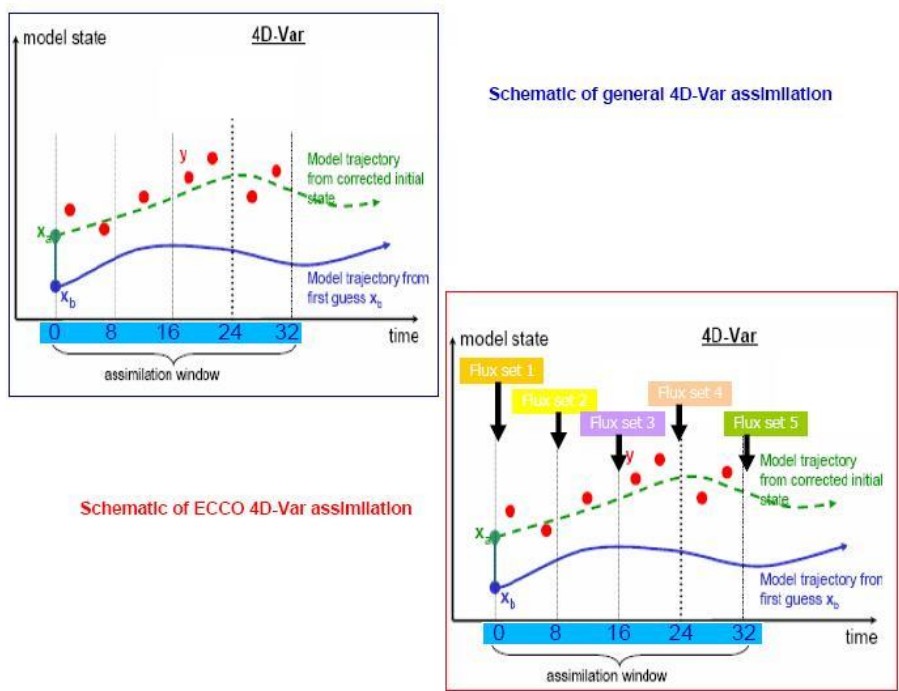

**Figure 2:** Schematics for the conventional 4D-Var with the initial model states as the control vector, and the ECCO-like 4D-

Var with both the initial model states and the external surface fluxes within the assimilation window as the control vector. Adapted from Singleton [2011].

Singleton [2011] conducted one experiment to compare the ocean analyses from the SC 4D-Var using the coupled model, and the ECCO-like 4D-Var using the ocean model forced by the atmosphere. The forced ocean model for the ECCO-



like 4D-Var is revised from the coupled Lorenz model [Peña and Kalnay, 2004], which now the ocean is forced by the external surface flux,

$$\dot{X} = \tau\sigma(Y - X) + f_X \tag{10}$$

$$\dot{Y} = \tau r X - \tau Y - \tau S X Z + f_Y \tag{11}$$

$$\dot{Z} = \tau S X Y - \tau b Z + f_Z \tag{12}$$

$$\dot{f}_X = 0 \tag{13}$$

$$\dot{f}_Y = 0 \tag{14}$$

$$\dot{f}_Z = 0 \tag{15}$$

The ECCO-like 4D-Var obtained its analysis $x_0^a$ by minimizing the cost function

$$J(x_0) = \frac{1}{2}[x_0 - x_0^b]^T B_0^{-1}[x_0 - x_0^b] + \frac{1}{2}\sum_{t=1}^{n}[H\left(M_{0,t}(x_0)\right) - y_t^o]^T R_t^{-1}[H\left(M_{0,t}(x_0)\right) - y_t^o] \tag{16}$$

where the control variable $x_0 = [X_0, Y_0, Z_0, f_{X,1}, f_{Y,1}, f_{Z,1}, f_{X,2}, f_{Y,2}, f_{Z,2}, \ldots, f_{X,n}, f_{Y,n}, f_{Z,n}]^T$ in ECCO-like 4D-Var includes both the initial ocean states $[X_0, Y_0, Z_0]^T$, and the constant surface fluxes $[f_{X,i}, f_{Y,i}, f_{Z,i}]$ that forces the ocean model for time-steps $1 + n \times (i - 1)$ to $n \times i$ for the $i$-th assimilation window. Here, $x_0^b$ represents the initial background states, $n$ is the length of an assimilation window, $H$ is an observation operator, $M_{0,t}$ is a forward operator from time $0$ to $t$, $y_t^o$ is an

observation vector at time $t$, and $R_t$ is an observation error covariance matrix. The background error covariance matrix **B**$_0$ is defined as

$$B_0 = \begin{bmatrix} B_{x,0} & 0 \\ 0 & B_f \end{bmatrix} \tag{17}$$

Where **B**$_{x,0}$ is the background error covariance of the initial ocean states estimated by the NMC methods. **B**$_f$ is the background error covariance for all the surface fluxes, which is assumed diagonal in our experiment, with its diagonal

elements representing the time-averaged variance of the flux estimates.

Running the ECCO-like 4D-Var requires the background of both initial ocean states and the surface fluxes (e.g., $f_{X,i}^b, f_{Y,i}^b, f_{Z,i}^b, k = 1, \ldots, n$) at all the time-steps. The real ECCO analysis system uses surface flux estimated from the NCEP Atmospheric Reanalysis [Kalnay et al., 1996] generated by an uncoupled atmospheric model forced by sea surface temperature. To get NCEP-like surface fluxes for our simple model, Singleton [2011] first replaced the active tropical ocean

with observations that are created from the true coupled trajectory in the coupled Peña and Kalnay [2004] model. Then the tropical atmosphere is forced by the ocean observations every 8 time-steps while it keeps a weak coupling with the extratropical atmosphere. A 10-member ETKF then produces the analyses for tropical and extratropical atmosphere every 8 time-steps. The final NCEP-like surface fluxes are calculated from the ensemble analysis mean of the tropical atmosphere (i.e., $[\overline{x_t^a}, \overline{y_t^a}, \overline{z_t^a}]$) through

$$f_X = -c\left(\overline{x_t^a} + k_2\right) \tag{18}$$





$$f_Y = c\left(\overline{y_t^a} + k_2\right) \qquad (19)$$

$$f_Z = -c_z \overline{z_t^a} \qquad (20)$$

For the assimilation experiment, ECCO-like 4D-Var integrates the ocean model forced by the constant NCEP-like surface fluxes every 8 time-steps. As the control experiment, Singleton [2011] includes one additional experiment which shares the same setting as the ECCO-like 4D-Var, except that its analyzed variables only include initial ocean states.

Figure 3 contrasts the performances of different 4D-Var approaches. For the forced ocean model, the ECCO-like 4D-Var approach that simultaneously estimates the ocean states and surface fluxes brings substantial improvements over the ordinary 4D-Var approach that only estimates the initial ocean states, with more significant improvement when utilizing a longer assimilation window. Both of these two 4D-Var analyses have the smallest error when adopting an assimilation window of 16 time-steps. However, the SC 4D-Var approach using the coupled model produces more accurate ocean analysis than the ECCO-like approach using the forced ocean model in terms of analysis RMSE. Besides, the error of SC 4D-Var ocean analyses keeps decreasing with longer assimilation window length up to 80 time-steps.

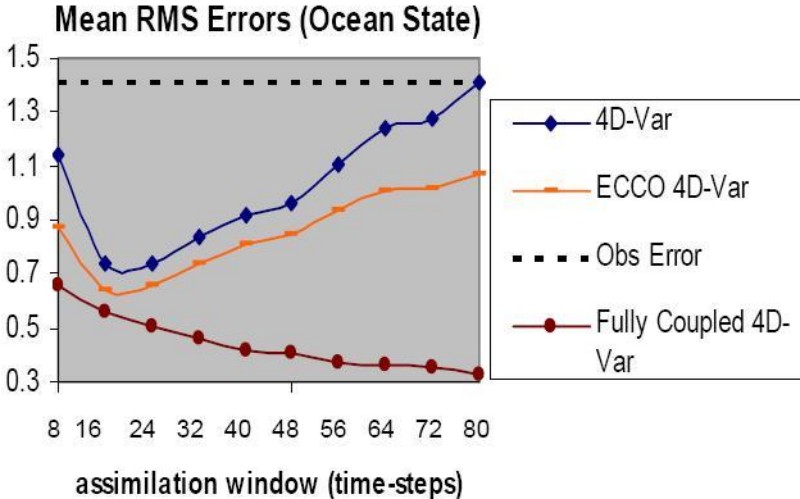

**Figure 3:** Time averaged analysis RMSE for conventional 4D-Var (blue) and the ECCO-like 4D-Var (orange) using the forced ocean model, and the SC 4D-Var (brown) using the fully coupled model. Adapted from Singleton [2011].

## 4. Comparisons of 3/4D-Var and EnKF in a coupled QG Model

We now discuss results by Penny et al. [2019] and Da [2022] who developed a CDA test bed using the coupled quasi-geostrophic (QG) atmosphere–ocean model MAOOAM (De Cruz et al., 2016), and compared the performance of 3/4D-Var and EnKF with different CDA strategies (i.e., UCDA, WCDA, quasi-SCDA, and SCDA). The MAOOAM model consists of a two-layer atmosphere and a single-layer ocean. It also includes Ekman dynamics at the atmosphere–ocean interface and the





simplified radiation parameterizations. To avoid interpretation complexity due to the inflation schemes in the EnKF, we set
the ensemble size as 40, greater than the total dimension of the model states (36), to avoid filter divergence without applying
the inflation schemes in the experiment.

Figure 4 (a)-(b) compares the atmosphere and ocean analyses by 3D-Var under three CDA strategies. Figure 4 (b)
shows that the WC and SC 3D-Var are more accurate than UC 3D-Var for ocean analyses. Increasing the frequency of
surface forcing exchange in UC 3D-Var reduces the ocean analysis error. However, the analysis error with a 6-hour forcing
update is still one order greater than the ocean analyses obtained from the coupled models. Among all three CDA
configurations, SCDA analyses are the most accurate for the coupled states.

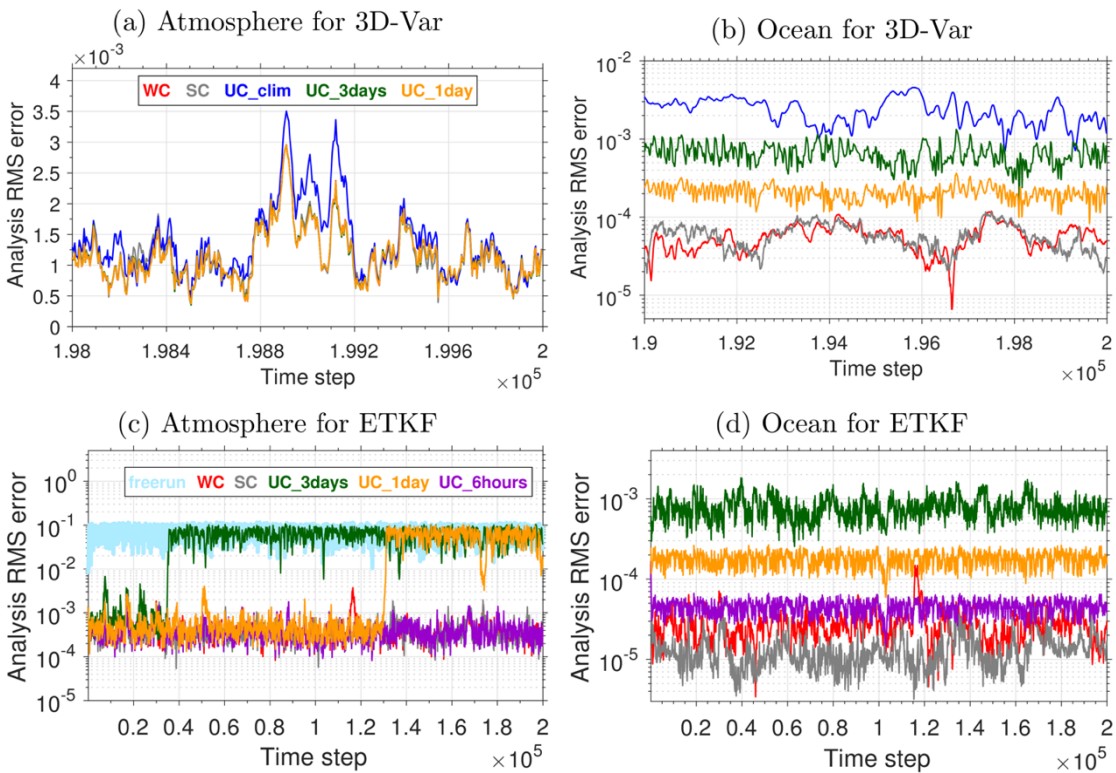

**Figure 4:** Panels (a)-(b): The analysis RMSE of the atmosphere and ocean for 3D-Var analysis with different CDA strategies
for the last 100 days for the atmosphere and last 500 days for the ocean. Panels (c)-(d) are similar to (a)-(b) except
for the ETKF during the whole experiment period (~27.4 year). Adapted from Penny et al. [2019] and Da [2022].

Similar to Figure 4 (a)-(b), Figure 4 (c)-(d) extends comparison to the ETKF. UC ETKF with forcing updated less
frequently than every 6 hours has filter divergence for the atmosphere, while such filter divergence does not occur for the
WC and SC ETKF that integrate the coupled models. This demonstrates the necessity of using coupled models for the



ensemble CDA systems. Similar to 3D-Var, switching from WC to SC ETKF reduces the analysis error for the coupled states. Besides, SC ETKF produces more accurate ocean analyses than the WC ETKF consistently, a feature not seen in the 3D-Var experiments. The improved ocean analyses by SC ETKF demonstrate one advantage of adopting an ensemble SCDA system.

Since comparisons of different CDA strategies show that the SCDA approach shows the most accurate analyses for both 3D-Var and ETKF, we now focus on evaluating the performance of SCDA under different observing networks and extending the comparison to 4D-Var and CERA-like variational analyses [Laloyaux et al., 2016]. The CERA-like variational system integrates the coupled model and generates incremental 4D-Var analysis for the atmosphere and 3D-FGAT analysis for the ocean using the outer loop coupling approach. Both 4D-Var and CERA adopt two outer loops in our experiments. For

the ETKF, the 40-member experiment uses no inflation, and the 20-member experiment uses multiplicative background error inflation of 1.01. Besides, all the assimilation methods adopt a 6-hour DA cycle.

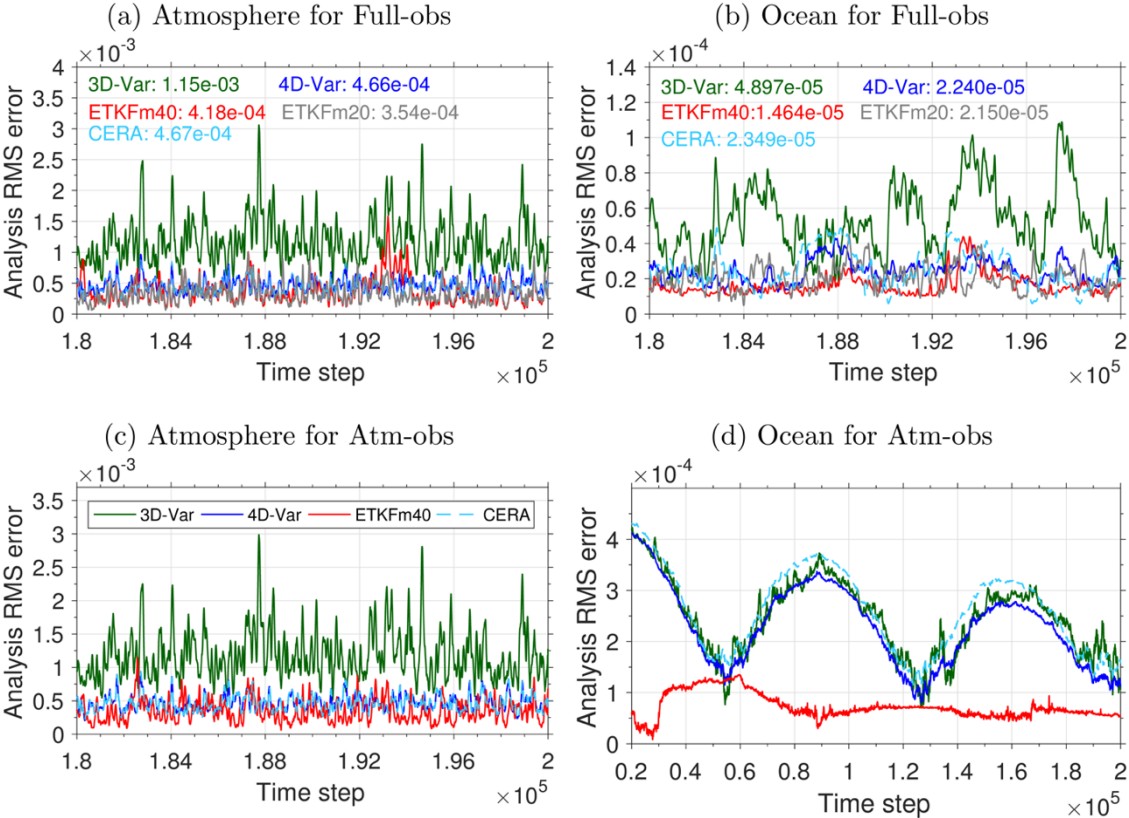

**Figure 5:** Panels (a)-(b): The analysis RMSE under full-coverage observing network for the atmosphere (left) and ocean

(right) with the SC 3D-Var (green), 4D-Var (blue), 4D-Var/3DFGAT CERA (cyan dash), 40-member ETKF (red), and 20-member ETKF (gray) for the last 1,000 days. Time averaged analysis RMSE for the last 13.7 years for all





methods are shown in the figure. Panels (c)-(d) are similar to (a)-(b) except for only assimilating atmosphere observations. Adapted from Penny et al. [2019] and Da [2022].

Figure 5 (a)-(b) shows that when observing both the atmosphere and ocean, the SC 40-member ETKF and 4D-Var have similar accuracies for the atmosphere and ocean analyses, lower than SC 3D-Var. The 20-member ETKF with inflation performs similarly to the 40-ember ETKF without inflation. For 4D-Var, applying more outer loops and longer assimilation window lengths further reduces the analysis error, consistent with the findings by Kalany et al. [2007] and Yang et al. [2012]. The CERA-like system with the outer loop coupling shows comparable performance as the SC 4D-Var and 40-

member ETKF in this scenario.

Figure 5 (c)-(d) compares the performance of different SCDA methods when only observing the atmosphere. For the atmosphere, ETKF, SC 4D-Var, and CERA present similar analysis accuracies smaller than SC 3D-Var. For the ocean, the SC ETKF stabilizes its analysis error after 10 years, while all variational data assimilation methods fail to stabilize the analysis error within the experiment period (~27.4 years). Interestingly, the CERA shows larger analysis errors among all

variational methods than the SC 3D-Var and 4D-Var, which utilize a coupled state background error in their formulations. This indicates that the outer loop coupling is insufficient to replace the role of a coupled-state background error covariance for variational CDA.

Though the CDA experiments with the coupled QG model indicate that the SC EnKF produces more accurate coupled analyses than the WC EnKF when the ensemble size is sufficient, it is unclear whether this conclusion still holds for

the real-observation experiments where the ensemble size is far less than the model dimension. Besides, the QG model mainly describes the midlatitude dynamics, while tropic dynamics is contributed significantly by convection, a mechanism not included in the QG model. Past studies [Kalnay et al., 1986; Peña et al. 2013; Ruiz-Barradas et al., 2017; Bach et al., 2019] have shown that the main driving force for the coupled atmosphere-ocean system differs in these two regions, with the ocean driving the atmosphere over tropics, and the atmosphere driving the ocean in mid-latitudes. It is necessary to examine

whether the conclusions drawn from the QG model can be applied to the Tropics.

## 5. SC EnKF with an intermediate complexity CGCM

In this section, we compared the performance of the SC and WC EnKF by conducting identical-twin experiments with an intermediate complexity CGCM, SPEEDY-NEMO [Sluka et al., 2016]. The CGCM SPEEDY-NEMO [Kucharski et

al., 2016] couples the atmospheric model Simplified Parameterizations, primitive-Equation Dynamics (SPEEDY) version 41 [Molteni, 2003; Kucharski et al., 2006], with the ocean model Nucleus for European Modeling of the Ocean (NEMO) version 3. The atmospheric model SPEEDY version 41 is a hydrostatic spectral model that solves primitive equations at a resolution of T30/L8. The ocean model NEMO adopts 30 vertical levels with z-coordinates and 2º tripolar grids that increase the resolution to 0.25º at the equator.



Sluka et al. [2016] implemented WC and SC EnKF systems for SPEEDY-NEMO by utilizing the existing separate EnKF systems SPEEDY-LETKF [Miyoshi, 2005] and Ocean-LETKF [Penny et al., 2011; 2013]. A 6-year perfect model OSSE is then conducted to compare the coupled-state analyses of the WC/SC EnKF. Both experiments use 40 members and adopt a 6-hour assimilation cycle for the atmosphere and oceans. Synthetic atmosphere observations are assimilated into the atmosphere in both experiments. Besides, the SCDA experiment assimilates those atmospheric observations into the ocean

while the WCDA experiment assimilates nothing into the ocean.

Figure 6 demonstrates that SC EnKF produces more accurate analyses of sea surface temperature and salinity than WC EnKF over the globe during the whole experiment period, with the most significant improvement in the midlatitude in the Northern Hemisphere. This analysis error reduction for the ocean temperature and salinity brought by SCDA also extends to the deep ocean layer (512m-2290m). Figure 7 examines the global map of analysis error reduction by SCDA for

the atmosphere and ocean. Overall, SCDA improves the analysis of the upper ocean temperature and salinity most significantly over the Tropics and the Northern Hemisphere. Interestingly, with no ocean observations assimilated into the atmosphere, the atmosphere analysis in the SCDA experiment still improves thanks to the more accurate ocean analysis through the coupled model integration.

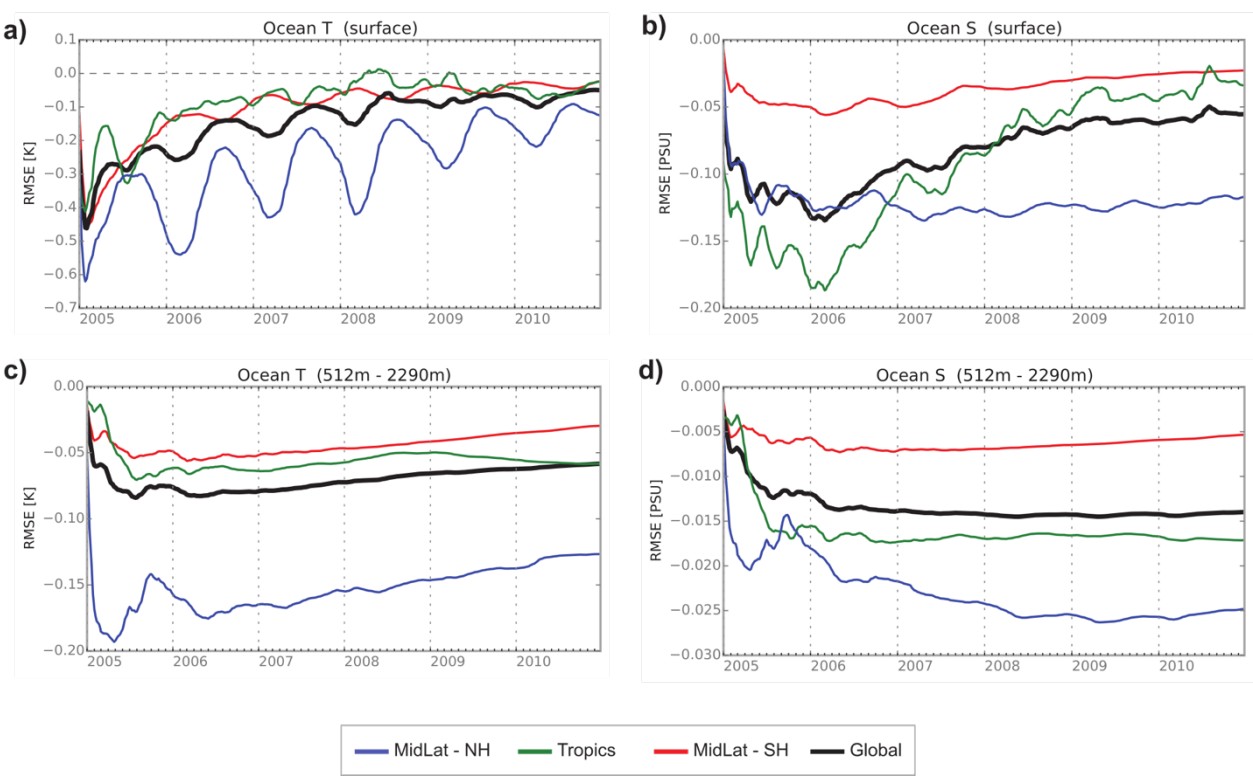


**Figure 6:** Spatially averaged difference of analysis RMSE with SCDA and WCDA for the ocean temperature and salinity at the surface (panels a-b), and at deep ocean (512m-2290m, panels c-d) in the Northern Hemisphere midlatitudes



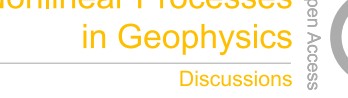

(blue), tropics (green), and Southern Hemisphere midlatitudes (red), and globe (black). Negative values mean
RMSE reduction by adopting SCDA. Adapted from Sluka et al. [2016] and Sluka [2018].


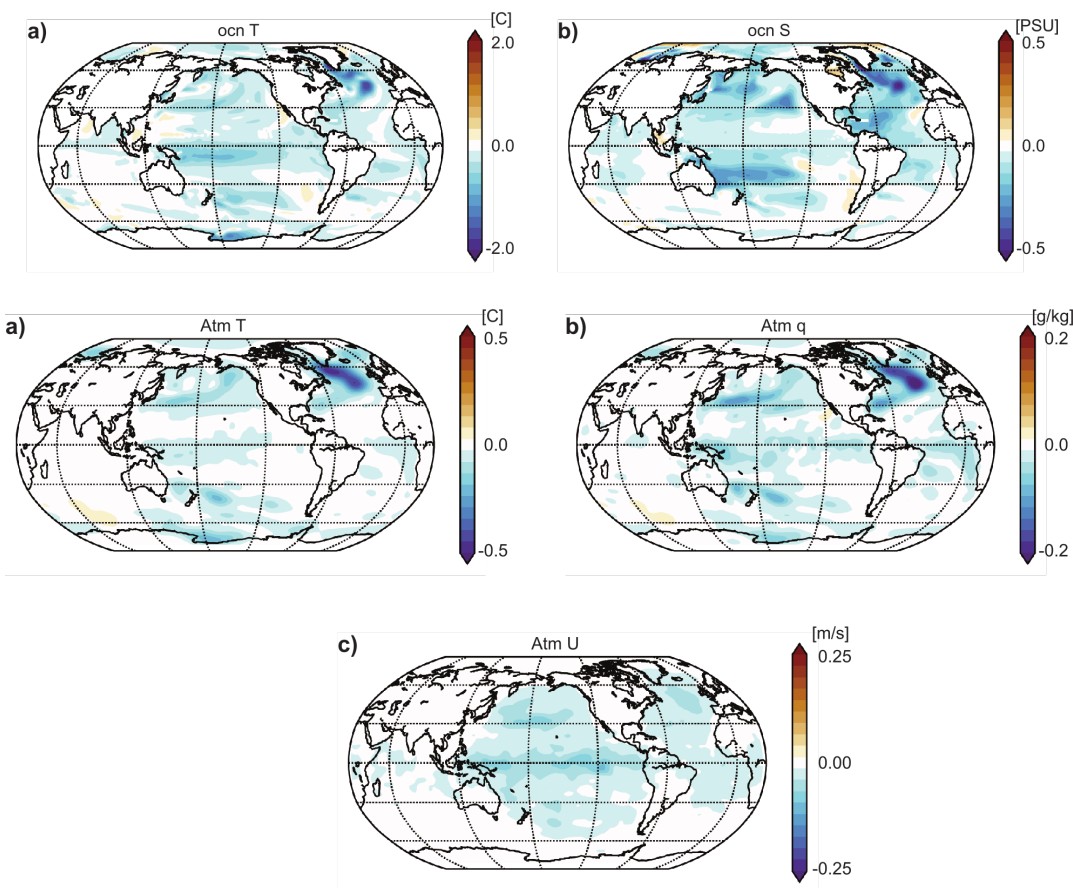

**Figure 7:** Time-averaged difference of analysis RMSE with SCDA and WCDA for the ocean surface temperature and
salinity (panels (a)-(b)), atmospheric temperature, humidity at the lowest atmospheric model level (panels (c)-(d)),
and the zonal wind speed throughout the troposphere (panel (e)) for the final 5 years (2006-2010) of the identical
twin experiment. Adapted from Sluka et al. [2016] and Sluka [2018].

## 6. SC EnKF with the state-of-the-art coupled model CFSv2

Sluka [2018] implemented a prototype WC and SC LETKF system CFSv2-LETKF for the operational coupled model
Climate Forecast System version 2 [CFSv2, Saha et al., 2006; 2014]. The atmospheric model Global Forecast System (GFS)
within the CFSv2-LETKF is a hydrostatic spectral model with hybrid pressure-sigma coordinates. It is configured with a
resolution of T62/L64 (~2 degrees). The ocean model GFDL Modular Ocean Model (MOM) version 4 is configured with 40
vertical levels using z* coordinates and tripolar horizontal grids of 0.5º that increase to 0.25º at the equator. The CFSv2



LETKF system was built upon the GFS-LETKF [Lien et al., 2016a; 2016b] and the MOM-LETKF [Penny et al., 2011; 2013], with many modules refactored so that the underlying software framework can be reused to implement WC and SC

EnKF systems for other coupled models. The CFSv2-LETKF is publicly available at https://github.com/UMD-AOSC/CFSv2-LETKF.

With the 50-member CFSv2-LETKF, Sluka [2018] conducted 3-month Observing System Experiments (OSEs) from June to August in 2015 to evaluate the benefits of SCDA over WCDA using real observations. The atmospheric model assimilates the same set of observations for both experiments (Table 3.1 in Sluka [2018]), while additional marine surface

reports are assimilated into the ocean model in the SCDA experiment. Unlike the SPEEDY-NEMO experiment, CFSv2-LETKF adopts a 6-hour assimilation cycle for the atmosphere and a 24-hour assimilation cycle for oceans to minimize the initial shock due to the frequent analysis update for the ocean.

Figure 8 shows that SCDA leads to reduced observation departures for the surface temperature observations than WCDA globally. Substantially improved observation fits is found in the Northern Hemisphere, with a misfit reduction of

13.1%, which is probably contributed by the dense marine surface reports in the Northern Hemisphere. In the Southern Hemisphere and over the Tropics, SCDA reduces the observation misfit by 3.8% and 2.1%, compared to WCDA.

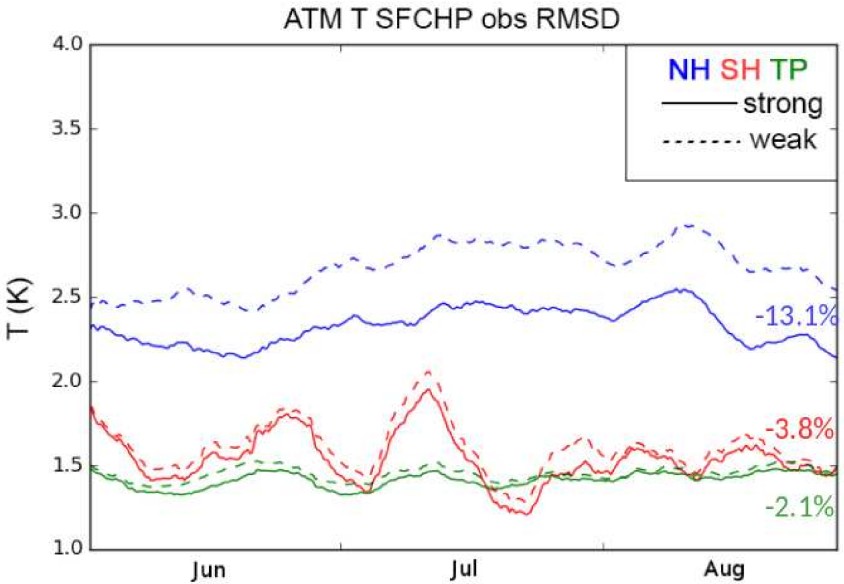

**Figure 8:** RMSD of observation minus 6-hour forecast (O-F) for atmospheric surface temperature observations with the SC (solid) and WC (dashed) CFSv2-LETKF over the Northern Hemisphere (NH), tropics (TR), and southern

Hemisphere (SH). Adapted from Sluka et al. (2016) and Sluka (2018).

Figure 9 verifies the model ocean temperature against independent ocean temperature profiles. SCDA shows better observation fitting than WCDA for the 100-m upper layers of tropical oceans. In the Northern Hemisphere, SCDA improves




the fitting for the 25-m upper layer but degrades the fitting below this depth. The degradation below 25-m depth is probably
due to the missing vertical localization not used in the Ocean-LETKF. With no vertical localization, the long-distance error
correlations between observations and analyzed variables cannot be reliably estimated by the small ensemble, especially for
the weak correlation from those physically "irrelevant" cross-domain state-observations pairs that appears more frequently in
the SCDA.

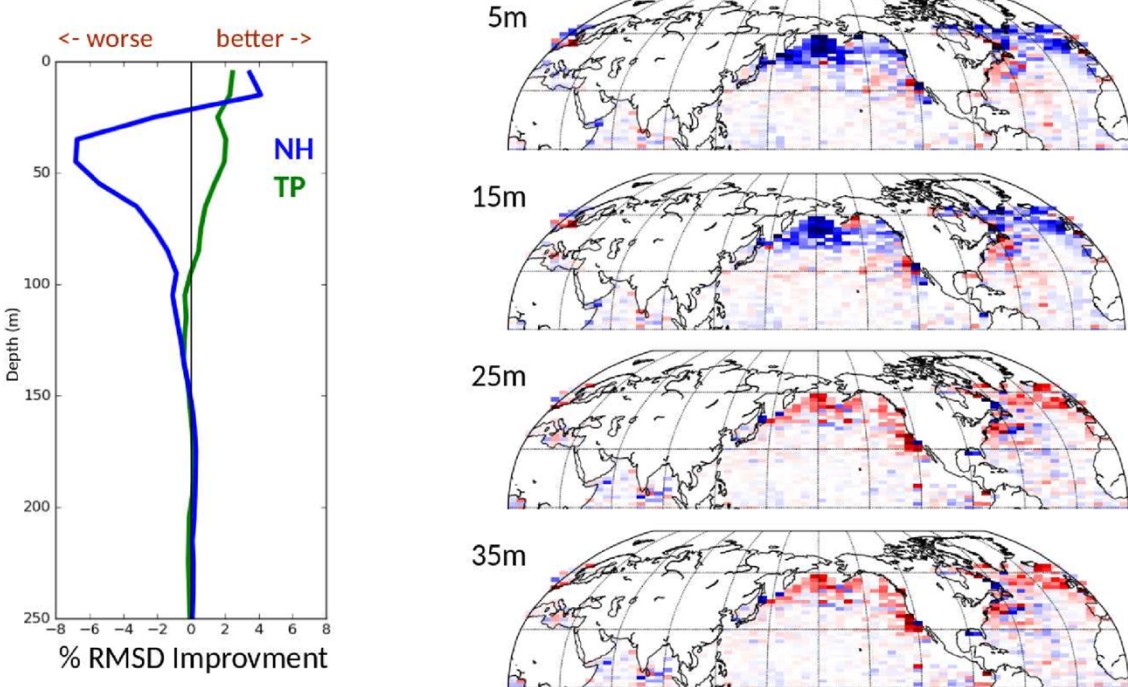

**Figure 9:** Difference of RMSD of observation minus 6-hour forecast (O-F) for ocean temperature with the SC (solid) and
WC (dashed) CFSv2-LETKF. The left panel shows the spatially averaged RMSD change that varies with the ocean
depth over the Northern Hemisphere (NH) and tropics (TR), and the right panel shows the spatial patterns of the
RMSD change at selected ocean depth. Adapted from Sluka et al. (2016) and Sluka (2018).

**7. Correlation-cutoff method for the SC EnKF**

Yoshida and Kalnay [2018] proposed the correlation-cutoff method, which can reduce the spurious error correlations among
different state-observation pairs, thus improving the performance of the SC EnKF with a small ensemble size. Through the
analysis of the Kalman Filter equations, Yoshida and Kalnay [2018] showed that the analysis increment due to the
assimilation of each observation is proportional to the square of the error correlations between the analyzed model state and
the observation simulations. In the correlation-cutoff method [Yoshida and Kalnay, 2018], only observations that show
strong time-averaged squared background error correlation with the model states are assimilated by the SC EnKF, since a
small ensemble cannot reliably estimate the weak error correlations for "irrelevant" state-observation pairs.



The underlying idea of the correlation-cutoff method is similar to the "variable localization" technique for the coupled atmosphere-carbon assimilation [Kang et al., 2011], in which the error correlation between physically irrelevant

variables (e.g., carbon flux and the specific humidity) are manually zeroed out for the EnKF. However, unlike the "variable localization" that removes the nonzero error correlation empirically, the correlation-cutoff method automates this process based on the time-averaged squared background error correlation using data acquired from offline assimilation experiments, which is desirable for CDA since it is nontrivial to determine whether the error correlation between cross-domain observation-state pairs should be zeroed out.


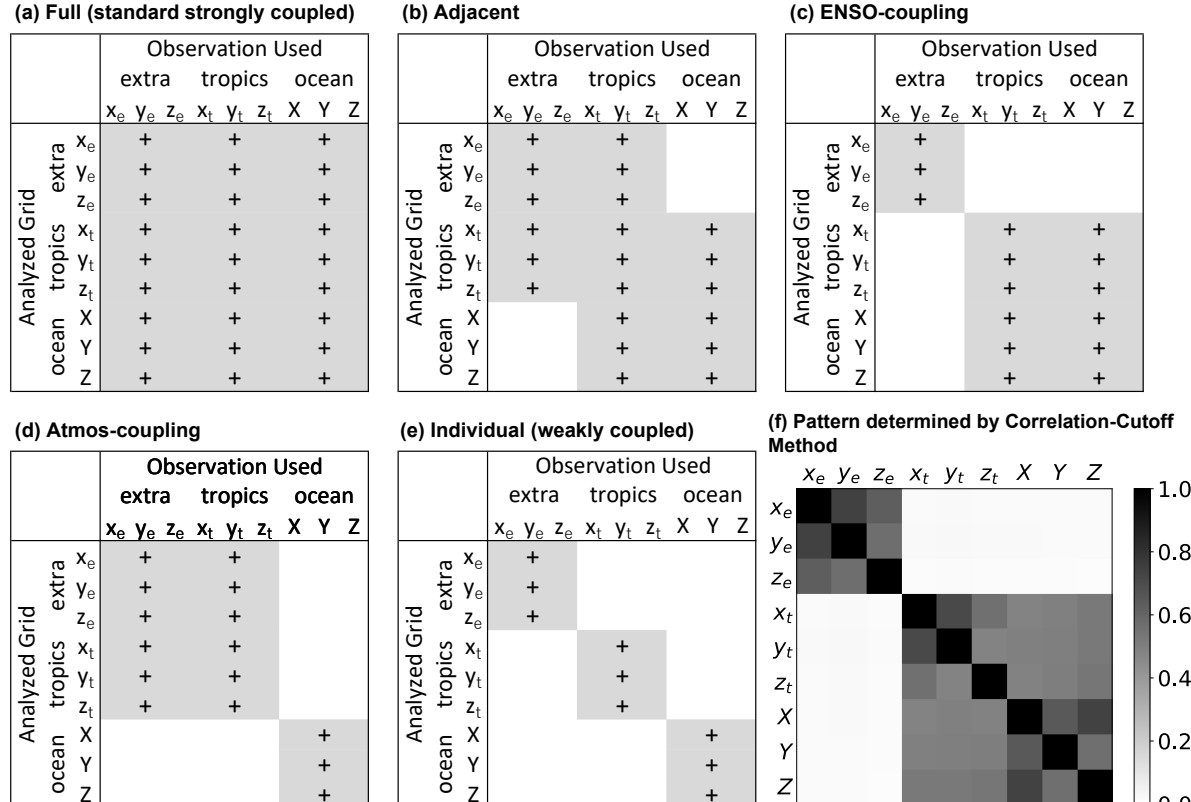

**Figure 10:** (a-e) Covariance localization patterns tested in the assimilation experiments of Yoshida and Kalnay (2018), and (f) the time-averaged squared background error correlation for different pairs of model state and observation types obtained from the independent offline LETKF experiments. Adapted from Yoshida and Kalnay [2018] and Yoshida

[2019].

Yoshida and Kalnay [2018] then examined the effectiveness of the correlation-cutoff method on SC EnKF using the coupled Lorenz system [Pena and Kalnay, 2004]. Figure 10 (f) shows that the localization pattern determined by the





correlation-cutoff method is like ENSO-coupling (Figure 10): with strong error correlation ($\overline{corr^2} \sim 0.5$) between the tropical

atmosphere and the tropical ocean, and weak correlation ($\overline{corr^2} < 0.03$) between the extra-tropical atmosphere and the other

two components. This squared correlation map suggests assimilating the extra-tropical observations into the extra-tropical

atmosphere, and tropical observations into the tropical atmosphere and ocean.

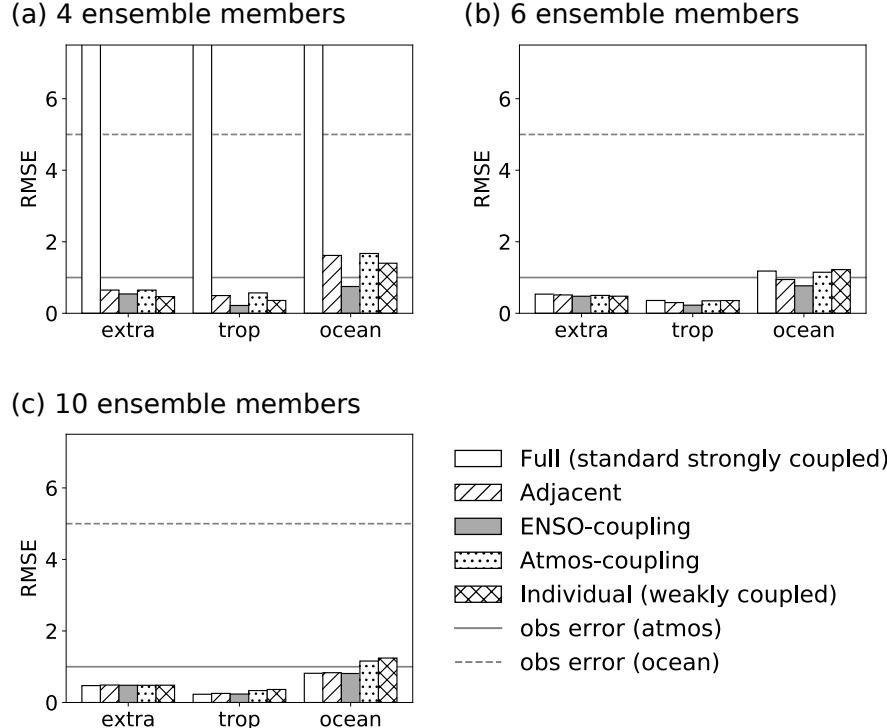

**Figure 11:** Time-averaged analysis RMSE with different localization patterns. Horizontal lines show the observation errors
for the atmosphere (solid) and ocean (dashed). Note that the filter diverged in the 4-member *Full* experiment.
Adapted from Yoshida and Kalnay [2018] and Yoshida [2019].

Figure 11 compares the analysis accuracy of the SCDA informed by the correlation-cutoff of the EnKF with five

different localization patterns (Figure 10), including WCDA, SCDA, and SCDA guided by the correlation-cutoff method.

All experiments are repeated with three different ensemble sizes of 4, 6, and 10. Figure 11 shows that SCDA ("Full"

experiment in the figure) is less accurate than WCDA ("Individual") or even experiences filter divergence with an

insufficient ensemble size of 4 or 6, while SCDA is more accurate than WCDA with a sufficient ensemble size of 10.

Meanwhile, SCDA guided by the correlation-cutoff method ("ENSO-coupling") generates the most accurate analysis

regardless of the ensemble size, demonstrating the necessity to ignore the weak error correlation for improved SC EnKF

analyses.





### 8. Emulate the localization functions with the neural networks

In this section, we discuss the results by Yoshida [2019] who applied the correlation-cutoff method to the more realistic models by using neural networks (NNs). Extending the correlation-cutoff method to a more realistic model is challenging because it requires functions that can predict the squared error correlations for each pair of observation and model state types and change values based on their spatial separation distance. For the operational SCDA application, this function must also be computationally cheap and fast since it is evaluated for all the observations within an influence radius around the analysis

grid. Yoshida [2019] proposed to train one neural network (NN) for each pair of observation and model state type that predicts the squared error correlation based on the attributes of the model state (e.g., geophysical location, time information) and observations (e.g., geophysical location and viewing geometry). Once trained, the NN can make fast predictions with low computational costs.

Yoshida [2019] first demonstrated the effectiveness of the NN in predicting the error correlation and its square by

using the NN to emulate the error correlations of four toy error correlation models under geostrophic balance. Predicting the error correlations instead of their squares is more challenging since the error correlation changes sign at different quadrants for error correlations of winds. The trained NNs will predict the error correlation with varying combinations of explanatory variables (up to 3) as inputs. The NN for each error correlation model is a two-layer feedforward NN with 10 hidden units, with the hyperbolic tangent chosen as the activation function. The training dataset is created by adding Gaussian error with a

standard deviation of 0.2 to the true error correlation. The trained NN is then obtained by minimizing the squared regression error with 1000 samples of the training datasets. Figure 12 shows that with proper explanatory variables (from the 2nd to the last columns) as the input, the NN can effectively predict the signs and values of the true error correlation (1st column). The other experiment that directly predicts the squared error correlation with the NNs shares similar results.

Yoshida [2019] then utilized the NN to predict vertical error correlations of the zonal wind for the intermediate

CGCM Fast Ocean Atmosphere Model (FOAM, Jacob [1997]). In this case, the NN is a two-layer feedforward NN with 30 hidden units, and it uses only 4 explanatory variables as its inputs: the distance between the analysis grid and the observation, the latitude of the analysis grid, and the vertical coordinate of the analysis grids and its counterpart for the observation. The NN is trained with the analysis ensemble from an offline 64-member WC ETKF experiment. Figure 13 shows that the error correlations predicted by the NN shares similar structures as those acquired using the NMC method

[Derber and Parrish, 1992], confirming that the NN can predict the error correlation for different state-observations pairs.

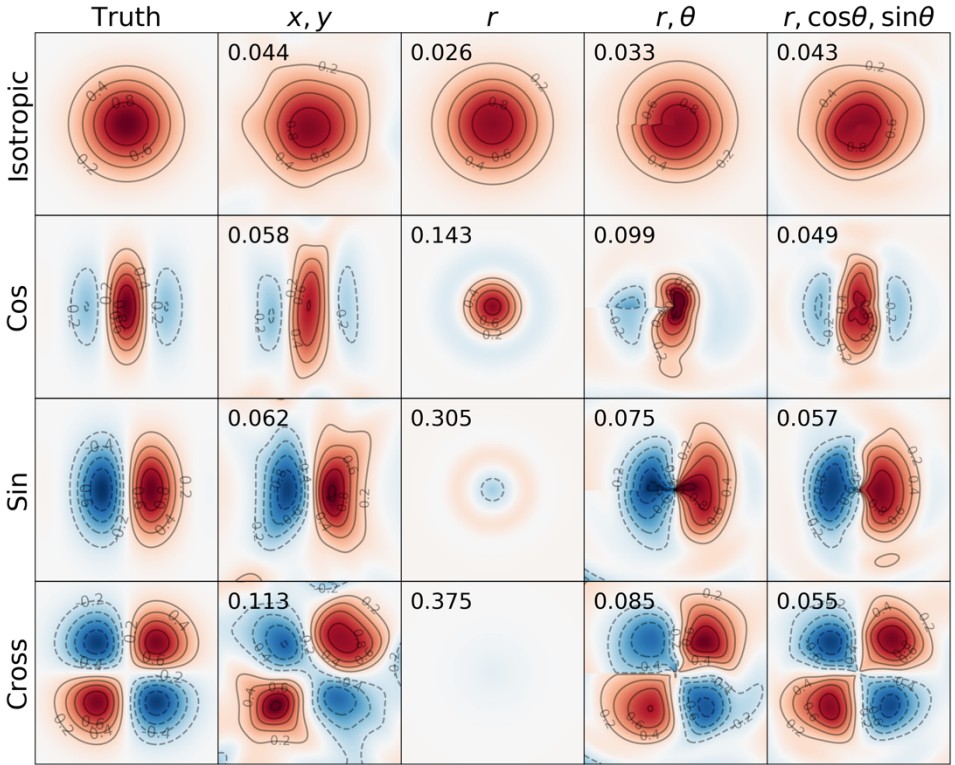

**Figure 12:** The true error correlations modeled by 4 toy correlation models (1st column) and the emulated ones with the neural network by adopting different sets of explanatory variables (columns 2-5). The RMS regression errors verified against independent validation datasets are shown in each panel. Adapted from Yoshida (2019).


With the error correlation square predicted by the NNs, the final localization value ρ informed by the correlation-cutoff method is calculated as

$$\rho = g(c) = \begin{cases} 0, & c^2 \leq c^2_{cutoff} \\ 1 - \left(\frac{1-c^2}{1-c^2_{cutoff}}\right)^2, & c^2_{cutoff} < c^2 \leq 1 \\ 1, & c^2 > 1 \end{cases} \quad (19)$$

where $c^2$ is the squared error correlation predicted by the NN, and $c^2_{cutoff}$ is a predefined cutoff parameter. A reasonable

$c^2_{cutoff}$ should be at least greater than 1/(ensemble size-1) since any correlation under this value is unreliable [Pitman, 1937]. For the later assimilation experiments, the cutoff parameter of 0.1 is selected.

Yoshida [2019] then conducted a 1-year OSSE with the coupled model FOAM to compare the performance of SC EnKF with the traditional localization functions and the localization function informed by the correlation-cutoff method with the NN. Figure 14 shows that the correlation-cutoff method with the NN improves the 24-hour forecast for different surface

atmospheric variables (i.e., surface pressure, temperature, humidity, and winds) almost everywhere, except at high latitudes




in the Northern Hemisphere, with the most significant improvements over the tropics. This improvement also extends to the upper atmosphere up to 250hPa. For oceans (Figure 15), the correlation-cutoff method improves the 24-hour forecast of sea surface temperature and salinity globally except at high latitudes in the Southern Hemisphere. Besides, the correlation-cutoff method also reduces the forecast error of ocean currents except at high latitudes in the Northern Hemisphere. Overall, the

correlation-cutoff method with the NN improves the analyses and forecasts of the SC EnKF.

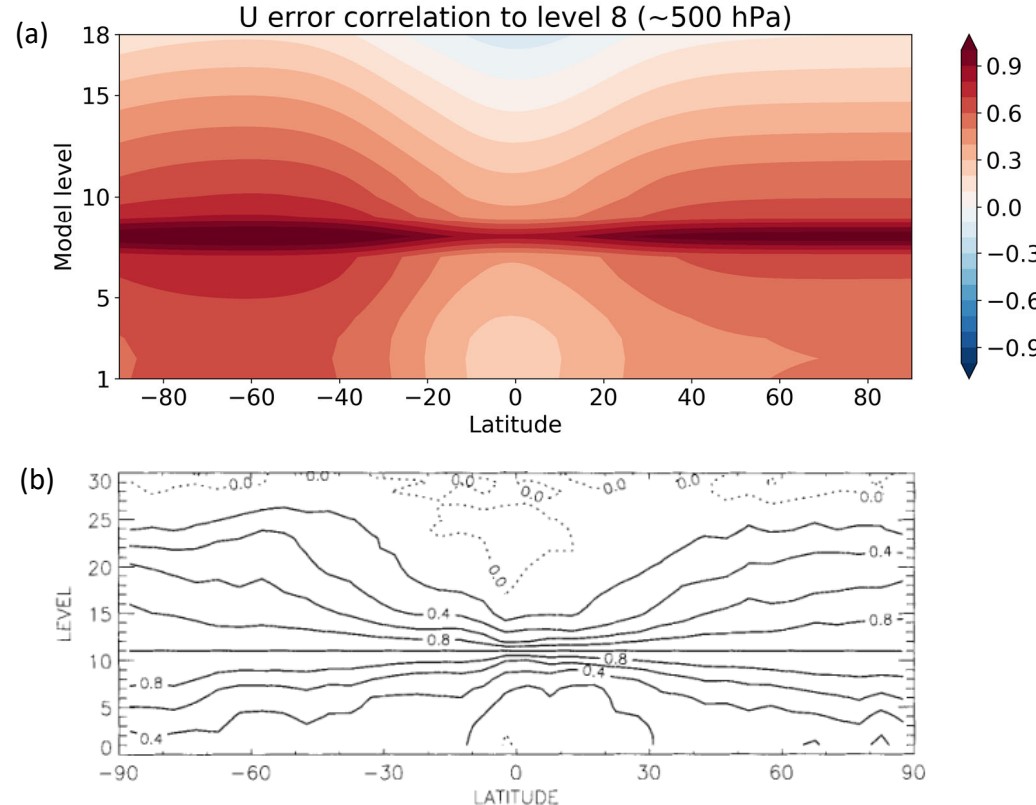

**Figure 13:** The vertical background ensemble auto-correlation of zonal winds to the model level of approximately 500hPa (a) emulated by the neural network for the model FOAM, and (b) calculated with the NMC method for the operational model by Ingleby in [2001]. Adapted from Yoshida [2019].


**Figure 14:** Difference of background (24-hour forecast) RMSEs between the correlation-cutoff with neural network and
standard strongly coupled EnKF OSSEs. Blue (red) colors show smaller (larger) errors in the correlation-cutoff
experiment. Errors are for atmospheric variables. Adapted from Yoshida [2019].

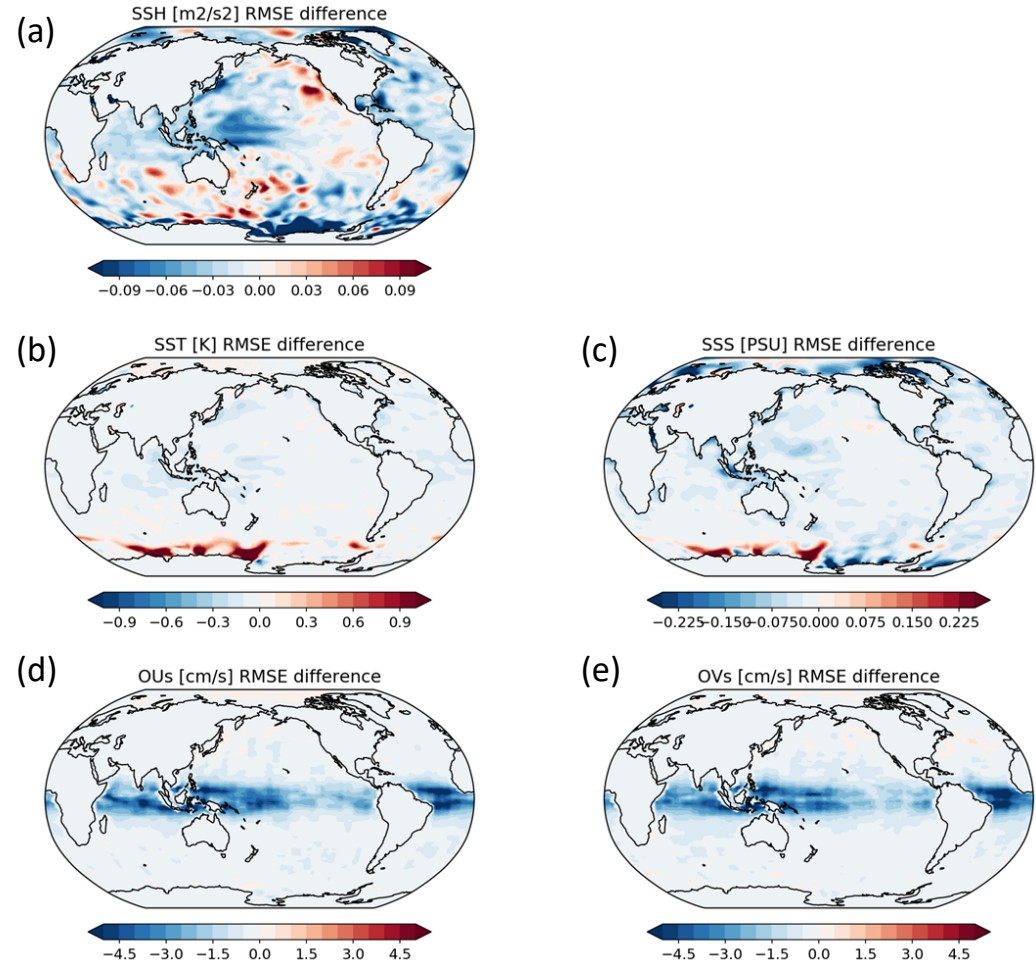

**Figure 15:** Same as Figure 14 but for oceanic variables. Adapted from Yoshida [2019].

## 9. Summary and discussion

We have reviewed our research progress about CDA by using a hierarchy of coupled models with increasing complexities, ranging from the simple coupled Lorenz model to the state-of-the-art operational coupled model CFSv2. With the Lorenz model, we proved that SC EnKF and 4D-Var could constrain the fast and slow modes of the coupled model simultaneously. EnKF produces the most accurate coupled analyses with a short assimilation window length. Applying 4D-extension or the Quasi-outer-loop allows the EnKF to utilize longer assimilation windows to improve the coupled analyses. Unlike EnKF, SC 4D-Var prefers long assimilation windows, consistent with the findings by Kalnay et al. [2007]. It is shown that the SC EnKF with a sufficient ensemble size and SC 4D-Var have similar accuracies if using their optimal assimilation window lengths. Compared to the ECCO-like 4D-Var with the forced ocean model, the SC 4D-Var using the coupled model can





produce more accurate ocean analysis, demonstrating the benefits of adopting the SCDA approach even for producing single-domain analysis.

Experiments with a coupled QG model show that SCDA produces more accurate analyses than WCDA and UCDA for both variational and ensemble methods. Besides, SC ETKF shows persistent smaller ocean analysis errors than WC ETKF, a phenomenon not observed for 3D-Var. Comparison of SCDA approaches under a full observing network shows that EnKF and 4D-Var reach similar analysis accuracy smaller than 3D-Var. The CERA-like approach using the "outer-loop coupling" shows comparable performance as the SC 4D-Var and ETKF. If only assimilating atmosphere observations, all variational assimilation methods using the static background error fail to stabilize their analyses for the experiment period, with the CERA-like system showing the worst performance, indicating that the outer-loop coupling approach alone cannot replace the role of the full-coupled background error covariance for the variational systems.

Given the similar performance of the SC 4D-Var and EnKF confirmed by the experiments with low-order models and the simple structure of the EnKF, we focused on developing EnKF-based CDA systems to which underlying software framework can be applied to the complex CGCM. Sluka et al. [2016] and Sluka [2018] developed a flexible LETKF-based CDA software framework and applied it to an intermediate-complexity coupled model SPEEDY-NEMO and the state-of-the-art operational coupled model CFSv2. Through assimilation experiments by assimilating synthetic or real atmospheric observations into the ocean through the SC EnKF with a small ensemble size, we found that SCDA produces more accurate lower atmosphere and upper ocean analyses than WCDA. However, we noticed that SCDA with the CFSv2-LETKF degrades the observation fits for the deep ocean layers, probably due to the suboptimal analysis update arising from the spurious error correlation estimated by the small ensemble used by the SCDA system.

Yoshida and Kalnay [2018] developed the correlation-cutoff method to alleviate the spurious error correlation problem in the SCDA. In the correlation-cutoff method, only those cross-domain observations that show strong ensemble correlations with the updated model variables are assimilated in the SCDA systems. Experiments with the coupled Lorenz model show that SCDA informed by the correlation-cutoff method outperforms the SCDA and WCDA regardless of ensemble size. To apply the correlation-cutoff method to complex CGCMs, Yoshida [2019] utilized the neural networks to acquire observation localization functions for different state-observation pairs systematically. The perfect model experiments with a CGCM showed promising results using this method.

As the computing resources increase, we expect SCDA with the EnKF to play a more critical role in producing coupled analyses. For now, the tremendous computational resources (i.e., long CPU runtime and related queue time, and high demand for disk storage) required by the EnKF-based SCDA systems prohibits the wide adoption of the EnKF-based CDA approaches. Efforts shall be made to reduce the computational resources related to CDA. For example, the online assimilation approach by Zhang et al. [2005, 2007] is an admirable attempt to alleviate this issue. Since their EnKF is implemented as a subroutine within the CGCM, all CDA procedures are conducted rapidly in the memory by avoiding frequent I/O of restart files. Other promising solutions include running the CGCM and its CDA package with reduced precisions [Váňa et al., 2017; Lang et al., 2021], and developing emulators for the CGCM using machine learning and

Artificial Intelligence techniques [Pathak et al., 2022; Lam et al., 2022]. Besides computational resource challenges, extending the SCDA approach to more coupled earth system components is also desirable. While our study has focused on coupled atmosphere-ocean analyses, the SCDA approach has shown its superiority to other CDA methods for other coupled components, such as coupled land-atmosphere DA [Lin and Pu, 2018; 2020].

Another potential future application for CDA is for coupled Earth–Human Systems, where Earth system
components are coupled with Human System components using bidirectional feedbacks [e.g., Motesharrei et al. 2014]. Dynamical models of the Human System are not yet broadly developed, leading to uncertainties when making projections using coupled models. CDA will be a crucial method to quantify and constrain these uncertainties [Motesharrei et al., 2016]. Furthermore, there are certain parameters of the Human System that could be reliably estimated from observations but there remain many uncertain parameters, especially coupling parameters. CDA can significantly contribute to estimation of these
parameters [e.g., Liu et al., 2014], especially when combined with Machine Learning algorithms. These advancements can help determine the Carrying Capacity of coupled human–natural systems and guide policymakers to keep these systems within their sustainable boundaries [Mote et al., 2020].

**Author Contributions**

EK supervised all the research by TS, TY, and CD. TS, TY, and CD conducted the CDA experiments and performed the analysis. SM and EK conducted research about the coupled Earth–Human Systems. All authors drafted and revised this manuscript together.

**Competing Interests**

The contact author has declared that none of the authors has any competing interests.

**Code Availability**

The source codes of the CFSv2-LETKF are available at https://github.com/UMD-AOSC/CFSv2-LETKF under GNU General Public License v2.0 (Last access: January 4, 2023).


**Acknowledgment**

CD was supported by the NASA Headquarters under the NASA Earth and Space Science Fellowship Program 80NSSC18K1403. CD also acknowledges support from NASA NNH20ZDA001N-MAP. TS was supported by the Monsoon Mission Directorate (grant MMSERPUnivMaryland). TY was supported by the Japanese Government Long-term Overseas
Fellowship Program. EK and SM were supported by the Monsoon Mission II funding for this work (Grant IITMMMIIUNIVMARYLANDUSA2018INT1) provided by the Ministry of Earth Science, Government of India. EK and SM were also supported by the CISESS grant "Advanced EFSO-Based QC Methods for Operational Use and Agile



Implementation of New Observing Systems", as a subaward from the National Oceanic and Atmospheric Administration
NOAA master grant NA14NES4320003.

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
