# Peer review of "Towards Strongly-coupled Ensemble Data Assimilation with Additional Improvements from Machine Learning"

_Nonlinear Processes in Geophysics, 2023_

## Referee Comment (RC2)

**Comment on the paper***:*
***"Towards Strongly-coupled Ensemble Data Assimilation with Additional Improvements from Machine Learning"***

General comments:

This paper compiles and comprehensively shows developments towards strongly coupled data assimilation. The studies included in this paper are organized according to the coupled model complexity, exploring the benefit of strongly coupled data assimilation over other approaches. They also include some studies in which one of the caveats of strongly coupled data assimilation is addressed by exploring low-order and intermediate-complexity coupled models. The last one is explored with the aid of neural networks. The paper is well-written and clear in general. However, it only focuses on the authors' research group developments and understanding of the field.

Although the title and the abstract are pertinent, the Authors should clearly state that the paper reviews their previous work. In its present form, the title indicates a novel contribution. At the same time, in being a review, the manuscript should also mention other key contributions to the field of coupled data assimilation.

Besides the above point, the paper is acceptable, with some technical corrections listed below.

The corrections are listed, in order of importance:
- The title and abstract should state clearly that this paper is a review (see general comment).
- In some of the experiments listed the details of how the DA is performed the variables assimilated are missing.
  1. Section 2: The SC ETKF and SC 4D-Var do not specify which variables are assimilated, and the components towards they are assimilated.
  2. Section 4: experiments do not explain the variables assimilated.
  3. Section 5: Missing which atmospheric variables are assimilated to atmosphere and ocean in both WCDA and SCDA.
- **Figure 1**: has a very poor quality and labels do not correspond to the experiments described in the manuscript, also the caption is confusing.
- In some results there is a confusion between high/low "accuracy" and low/large "error". This, in lines:
  1. **ln 255 - 256:** "… the SC 40-member ETKF and 4D-Var have similar accuracies for the atmosphere and ocean analyses, lower than SC 3D-Var." Change "lower" for "higher".
  2. **ln 262:** "…ETFK SC 4D-Var and CERA present similar analysis accuracies smaller than SC 3D-Var." This statement is wrong: the accuracy of 3D-Var is lower than for the other experiments.
  3. **ln 471:** "… EnKF and 4D-Var reach similar analysis accuracy smaller than 3D-Var." This statement is wrong: 3D-Var accuracy is smaller than EnKF and 4D-Var.
- **Figure 9:** The caption mentions lines that are not included in the figure. This is: the dashed lines for WCDA experiments, for the left panel. Besides, there is no indication on the meaning of the colors. Red=reduction, Blue=increase?
- **ln 215:** There is not an explanation on what or which are the "quasi-SCDA" methods.

- **ln 222 - 224:** They talk about an experiment that uses 6-hour forcing in Figure 4 (a)-(b), but these subfigures do not present such experiment.
- **Figure 7:** The labels of the subfigures are wrong. Panels (c)-(d) do not exist.
- **ln 338:** It is mentioned the result for WCDA experiment, but the figure does not show it, or it is not clear.

Technical corrections:
- Make sure all the captions have the same style for the citations.
- **ln 63**: Citation for the 3D-FGAT method is needed.
- **ln 104:** "… Kalnay [2004], of which equations are written as." Replace the period with a colon.
- **ln 123:** "…standard deviation of $\sqrt{2}$. Besides, Assimilation experiments…" Change capital to low-case in word 'Assimilation'.
- **ln 124:** "… experiments with the Ensemble Transform Kalman Filter (ETKF) in this section uses 9 members." Mismatch between subject and verb. Change "uses" to "use".
- **ln 177:** "…, and constant surface fluxes $[f_{X,i},\ f_{Y,i},\ f_{Z,i}]$ that forces …" Change "forces" to "force",
- **ln 178:** "… $x_0^b$ represents the initial background states, …" Change "states" for "state".
- **ln 183:** NMC is an acronym not defined.
- **ln 225:** "… update is still one order greater than… " Change to "… update is still  one order of magnitude greater than… "
- **ln 287:** Acronym OSSE is not defined.
- **ln 313:** missing indent at beginning of paragraph.

---

## Author Comment (AC1)

Review for "Towards Strongly-coupled Ensemble Data Assimilation with Additional Improvements from Machine Learning" by Kalnay et al.

This manuscript reviews the coupled data assimilation and strongly coupled data assimilation research conducted by Dr. Kalnay's group. The manuscript is well-written, with appropriate literature references. The content is very beneficial for the data assimilation community as well as the general readers who are interested in coupled data assimilation. I think it is well fit for NPG. In terms of the manuscript, I only have a few minor comments.

**R:** Thank you very much for your comments. All the line numbers in our response refers to the line numbers in the manuscript with tracked changes.

1. I can understand the abbr. SC and WC represent strongly-coupled and weakly-coupled, respectively. Please define them in the manuscript.

**R:** We have added definitions of uncoupled data assimilation (UCDA), weakly coupled data assimilation (WCDA), and strongly-coupled data assimilation (SCDA) in the first paragraph of Section 1, Introduction:

- L44: … the *uncoupled data assimilation (UCDA)* approach, which obtains independent analyses of different Earth system components based on the forecasts from uncoupled models, …
- L47: the *weakly coupled data assimilation (WCDA)* approach by creating separate analyses of the atmosphere and oceans, assimilating their domain observations based on the forecasts initialized from a coupled model.
- L52: the *strongly-coupled data assimilation (SCDA)* approach, which creates coupled analyses by assimilating the same set of the all-domain observations into different Earth system components,…

We also defined the quasi-SCDA in the second paragraph of Section 1, Introduction:

- L65: … implemented a *Quasi-SCDA* system through the "outer loop coupling", where the incremental 4D-Var atmospheric and 3D-FGAT oceanic analyses share the same outer loops so that their updated analyses will be used together to acquire the new model trajectory for the next round…

To ensure these definitions are clearly presented, we made those key words *italic* in their definition, and added the following sentence to L43:

"Different CDA strategies have been developed and summarized in Penny et al. [2017]"

And revised the section (in bold) related to quasi-SCDA from L65:

"The European Centre for Medium-Range Weather Forecasts (ECMWF) implemented the "outer loop coupling", where the incremental 4D-Var atmospheric and **3D-Var with the First Guess at the Appropriate Time (3D-FGAT, Lee et al., [2004]; Lawless [2010])** oceanic analyses share the same outer loops so that their updated analyses will be used together to acquire the new model trajectory for the next round [Laloyaux et al., 2016; 2018]. **Though cross-domain observations are not directly assimilated into separate earth components, separate earth component analyses benefit from a more coherent coupled-state through dynamical coupling at the data**

**assimilation step. Based on Penny et al. [2017], outer loop coupling belongs to the Quasi-SCDA methods.**"

Reference:

Penny, S., Akella, S., Alves, O., Bishop, C., Buehner, M., Chevallier, M., Counillon, F., Draper, C., Frolov, S., and Fujii, Y.: Coupled Data Assimilation for Integrated Earth System Analysis and Prediction: Goals, Challenges and Recommendations. World Meteorological Organization, WWRP 2017-3, 50, URL https://www.wmo.int/ pages/prog/arep/wwrp/new/documents/Final WWRP 2017 3 27 July.pdf.

The same comment also applies to abbr. "NMC"
**R:** NMC method refers to "National Meteorological Center" method [Parrish and Derber, 1992]. We have revised L200: "Where $B_{x,0}$ is the background error covariance of the initial ocean states estimated by the NMC methods"
to
"Where $B_{x,0}$ is the background error covariance of the initial ocean states estimated by the **National Meteorological Center (NMC) method [Parrish and Derber, 1992].**"

We also added the following reference:
Parrish, D. F, and Derber J.C.: The National Meteorological Center's spectral statistical interpolation analysis system. *Mon. Weather Rev.,* 120:1747-1763.

Ln 11-12: the simple coupled Lorenz model —> a simple coupled Lorenz model
**R:** Corrected.
L11-12 "…, ranging from a simple coupled Lorenz model to …"

2. Ln 19: I am confused about the "full-rank" EnKF
**R:** The "full-rank" EnKF refers to the EnKF where we use an ensemble size greater than the size of analyzed variables. We use the full-rank EnKF in this experiment so that we don't need to apply any inflation or relaxation methods to maintain the ensemble spread. This simplifies our interpretation of experiment results.

3. Ln 24 55 upper oceans —> upper ocean
**R:** Corrected.
L24: "of the atmosphere and upper ocean"

4. Ln 125-126, The smallest RMSE shows at an assimilation interval of 8 time-steps that is your smallest assimilation interval. I think it is worth pointing out here.
**R:** Thanks for this comment. We added this point to the discussion about the EnKF results of the Lorenz model in L137:

"Singleton [2011] found that SC ETKF has the smallest analysis Root Mean Square Error (RMSE) when adopting an assimilation interval of 8 time-steps, **which is the smallest assimilation interval used in that study."**

5. Fig. 2  From my understanding, ECCO only updates the boundary forcing and parameters, not ocean state variables.  From the figure, the initial conditions of model states are updated by DA. Please clarify it.

**R:** In our study, the ECCO-like system updates both the initial conditions and the surface forcing. This is to mimic the ECCO approach documented in Stammer (2004), Page 4, paragraph 15: "In the present calculation, the control vector includes the **three-dimensional initial condition potential temperature, $\theta$, and salinity, S, fields, as well as the daily surface forcing fields of net heat, net freshwater, and momentum fluxes** over the full 10 years."

6. Comparisons of 3/4D-Var and EnKF in a coupled QG Model

There are UC_clim, UC_3days and UC_1day.  it is worth providing details on their adjustment. Which is equivalent to the regular UC applying in the atmosphere and ocean?

**R:** The intention of using a different forcing update for the UCDA is to investigate the benefits of using uncoupled and coupled models. As we can see in Figure (a)-(b), increasing the forcing update frequency for the uncoupled model reduces the analysis RMSE for both the atmosphere and ocean.

For the UC 3D-Var, we chose the slowest forcing update as 1 day to mimic the common approach adopted by operational centers where their uncoupled atmosphere model uses the daily SST products as the surface forcing.

The improved ocean does not enhance the RMSE in the atmosphere throng dynamic coupling, which needs some discussion.

**R:** Based on our results, we cannot draw the conclusion that "The improved ocean does not enhance the RMSE in the atmosphere through dynamic coupling". Based on our calculation of the averaged analysis RMSE (scale factor 10^-5) over the last ~11 years, the SCDA atmosphere shows slightly better analysis than the WCDA atmosphere.

|  | WCDA | SCDA |
|---|---|---|
| Atmosphere | 116.0 | **115.9** |
| Ocean | 5.516 | **4.915** |

Ln 225-226 Fig. 4(a-b) only demonstrates two CDAs (WC, SC), not three methods,

**R:** In Figure 4(a-b), we showed WC (red), SC (gray) and UCDA with three different forcing update frequencies (climatology in blue, 3 days in green, and 1 day in yellow).

From fig. 4b, I can not conclude SC is better than WC.

**R:** Thanks for this comment. The conclusion that SC 3D-Var is better than WC 3D-Var is based on the averaged analysis RMSE (scale factor 10^-5) over the last ~11 years

|  | WCDA | SCDA |
|---|---|---|

| Atmosphere | 116.0 | **115.9** |
|------------|-------|-----------|
| Ocean | 5.516 | **4.915** |

Besides, we also find that for the ocean analysis, the SC 3D-Var (gray) shows smaller RMSE than the WC 3D-Var (red) during the spin-up period, as shown below.

[Figure]

7. Ln 255 "lower than SC 3D-Var" should be "higher than SC 3D-Var"
**R:** Corrected.

L280: "Figure 5 (a)-(b) shows that when observing both the atmosphere and ocean, the SC 40-member ETKF and 4D-Var have similar accuracies for the atmosphere and ocean analyses, **higher than SC 3D-Var**."

8. Ln 257 "For 4D-Var, applying more outer loops and longer assimilation window lengths further reduces the analysis error". The state mentioned here has no support.
**R:** We conducted additional 4D-Var experiments with a 12-hour assimilation window and up to 4 outer-loops, and results show that they slightly reduce RMSE. Since those results are not new findings and have been shown in Kalnay et al. [2007] and Yang et al, [2012], we do not show the figures here.
We revise this sentence in L281 as:
"For 4D-Var, applying more outer loops (i.e., 3 and 4) and longer assimilation window lengths (i.e., 12 hours) further reduces the analysis error (figures not shown here), …"

9. "accuracies smaller than SC 3D-Var" should be "higher than"
**R:** Corrected.
L287: "For the atmosphere, ETKF, SC 4D-Var, and CERA present similar analysis accuracies **higher** than SC 3D-Var."

10. Fig.6 shows that the different RMSE between SC and WC has not reached to equivalent, especially the surface T/S, which needs to point out.

**R**: we add the following discussion to L325: "Longer model integration is needed to evaluate the performance of the SC and WC EnKF after the ocean surface temperature and salinity finishes spin-up."

11. Ln 340 it is confusing for the statement "due to the missing vertical localization not used in the Ocean-LETKF." Please rephrase.
**R:** We rephrase the sentence in L367 "The degradation below 25-m depth is probably due to the missing vertical localization not used in the Ocean-LETKF."

as

"Since no vertical localization is applied in the ocean LETKF update, the degradation below 25m-depth is probably due to the sampling error caused by the small ensemble size."

12. Ln 356 "states are assimilated by the SC EnKF". The mean of SC EnKF here indicates the cross-model update, which is different from the other places indicating the whole DA algorithm.
**R:** Yes, we are performing cross-model update here, and the correlation-cutoff method [Yoshida and Kalnay, 2018] are designed specifically for the strongly-coupled EnKF. Note in our Figure 11, we also have the Standard SCDA (corresponding to the "Full" Pattern as shown in Figure 10), and it shows larger RMSE than the SCDA with the correlation cutoff method (corresponding to panel (c) and (f) in Figure 10).

---

## Author Comment (AC2)

**Comment on the paper*:***
***"Towards Strongly-coupled Ensemble Data Assimilation with Additional Improvements***
***from Machine Learning"***

General comments:
This paper compiles and comprehensively shows developments towards strongly coupled data assimilation. The studies included in this paper are organized according to the coupled model complexity, exploring the benefit of strongly coupled data assimilation over other approaches. They also include some studies in which one of the caveats of strongly coupled data assimilation is addressed by exploring low-order and intermediate-complexity coupled models. The last one is explored with the aid of neural networks. The paper is well- written and clear in general. However, it only focuses on the authors' research group developments and understanding of the field.

Thank you very much for your comments. All the line numbers in our response refers to the line numbers in the manuscript with tracked changes.

Although the title and the abstract are pertinent, the Authors should clearly state that the paper reviews their previous work. In its present form, the title indicates a novel contribution.

**R:** Thank you for this suggestion. We have revised the original title "Towards Strongly-coupled Ensemble Data Assimilation with Additional Improvements from Machine Learning"

to

"Review Article: Towards Strongly-coupled Ensemble Data Assimilation with Additional Improvements from Machine Learning"

At the same time, in being a review, the manuscript should also mention other key contributions to the field of coupled data assimilation.

**R:** In the Introduction Section (L39-L92), we have mentioned numerous important contributions in both variational and ensemble coupled data assimilation. We also mentioned two other excellent workshop or review papers by Penny et al. [2017] and Zhang et al. [2020].

We made the following revisions to narrow down the scope of our paper:
- We emphases that we focus on 1) model state estimations and 2) investigating the impacts of coupled atmosphere-ocean data assimilation on the coupled analysis and short-range weather forecasts.
- We refer to the review paper by Zhang et al. [2020] to the readers who are interested in the parameter estimation using coupled model and other coupled DA applications.

We add the following discussion (in Bold) to L93:

"…with increasing complexities. **We focus on model state estimations and impact of atmosphere-ocean CDA on coupled analysis and short-range weather forecast. Besides model state estimation, Zhang et al. [2020] recently reviewed parameter estimations and other important applications of CDA.**"

Reference:
Zhang, S., Liu, Z., Zhang, X., Wu, X., Han, G., Zhao, Y., Yu, X., Liu, C., Liu, Y., Wu, S., Lu, F., Li, M., and Deng, X.: Coupled data assimilation and parameter estimation in coupled ocean–atmosphere models: a review, Climate Dynamics, 54, 5127–5144, https://doi.org/10.1007/s00382-020-05275-6, 2020.

Besides the above point, the paper is acceptable, with some technical corrections listed below. The corrections are listed, in order of importance:
- The title and abstract should state clearly that this paper is a review (see general comment).

**R:** We have revised the title as "**Review Article**: Towards Strongly-coupled Ensemble Data Assimilation with Additional Improvements from Machine Learning"

- In some of the experiments listed the details of how the DA is performed the variables assimilated are missing.
  1. Section 2: The SC ETKF and SC 4D-Var do not specify which variables are assimilated, and the components towards they are assimilated

**R:** The analyzed variables are the full 9-element state vector of the Pena and Kalany [2004] model, and the synthetic observations are created for all these 9 elements.

We make the following addition (in bold) to L131 as:
"Singleton [2011] obtained the nature run by integrating the model using the 4th-order Runge-Kutta method with a time step $\Delta t = 0.01$. **The analyzed variables in the data assimilation experiments are the full 9-element state vector.** Observations are generated every 8 time-steps by adding to the true **9-variable** model states the uncorrelated Gaussian errors with zero mean and a standard deviation of $\sqrt{2}$."

  2. Section 4: experiments do not explain the variables assimilated.

**R:** The analyzed variables are the 36 nondimensionalized coefficients of spectral modes for the atmosphere ($N_a$=20) and ocean ($N_o$=16). Synthetic observations are created from the full true state vectors plus 10% natural variability. We add the following sentence to L238 as

"It also includes Ekman dynamics at the atmosphere–ocean interface and the simplified radiation parameterizations. **The analyzed variables are the 36 nondimensionalized coefficients of spectral modes for the atmosphere ($N_a$=20) and ocean ($N_o$=16).**"

We also added the following sentence to L242 as
"Figure 4 (a)-(b) compares the atmosphere and ocean analyses by 3D-Var under three CDA strategies. **Each experiment assimilates the synthetic observations of the full state vector.**"

3. Section 5: Missing which atmospheric variables are assimilated to atmosphere and ocean in both WCDA and SCDA.

**R:** The atmospheric observations in Section 5 include: surface pressure, vertical profile of temperature, humidity, zonal and meridional winds. We revise L312 as:

". Synthetic atmosphere observations **(i.e., surface pressure, vertical profile of temperature, humidity, zonal and meridional winds)** are assimilated into the atmosphere in both experiments."

- **Figure 1**: has a very poor quality and labels do not correspond to the experiments described in the manuscript, also the caption is confusing.

**R:** We have revised the captions of Figure 1 as

"**Figure 1:** Time-averaged Analysis RMSE for SC 4D-Var (green), SC ETKF-QOL (red), SC ETKF with the "atmos-coupling" (Figure 2 of Yoshida and Kalnay [2018]) as the localization pattern (cyan) and its 4D extension (blue) for the extratropical atmosphere (top left), tropical atmosphere (top right), and ocean (bottom). Adapted from Singleton [2011]."

- In some results there is a confusion between high/low "accuracy" and low/large "error". This, in lines:
  1. ln 255 - 256: "... the SC 40-member ETKF and 4D-Var have similar accuracies for the atmosphere and ocean analyses, lower than SC 3D-Var." Change "lower" for "higher".

  **R:** corrected.

  2. ln 262: "...ETFK SC 4D-Var and CERA present similar analysis accuracies smaller than SC 3D-Var." This statement is wrong: the accuracy of 3D-Var is lower than for the other experiments.

  **R:** corrected.

  3. ln 471: "... EnKF and 4D-Var reach similar analysis accuracy smaller than 3D-Var." This statement is wrong: 3D-Var accuracy is smaller than EnKF and 4D-Var.

  **R:** corrected.

- **Figure 9:** The caption mentions lines that are not included in the figure. This is: the dashed lines for WCDA experiments, for the left panel. Besides, there is no indication on the meaning of the colors. Red=reduction, Blue=increase?

**R:** We have revised the caption of Figure 9 as:

"**Figure 9:** RMSD reduction of observation minus 6-hour forecast (O-F) for ocean temperature by switching from WC to SC CFSv2-LETKF. The left panel shows the spatially averaged RMSD change (improvements with positive value, and degradation with negative value) that varies with the ocean depth over the Northern Hemisphere (NH, blue) and tropics (TR, green). The right panel shows the spatial distribution of the RMSD by switching

from WC to SC (improvements in blue, and degradation in red) at selected ocean depth. Adapted from Sluka et al. [2016] and Sluka [2018]."

- **ln 215:** There is not an explanation on what or which are the "quasi-SCDA" methods.

**R:** The quasi-SCDA method first appears at L65:

"The European Centre for Medium-Range Weather Forecasts (ECMWF) implemented a *Quasi-SCDA* system **through the "outer loop coupling", where the incremental 4D-Var atmospheric and 3D-Var with the First Guess at the Appropriate Time (3D-FGAT, Lee et al., [2004]; Lawless [2010]) oceanic analyses share the same outer loops so that their updated analyses will be used together to acquire the new model trajectory for the next round** [Laloyaux et al., 2016; 2018]."

Compared to the WCDA, the Quasi-SCDA method such as outer loop coupling still performs separate analysis update using their own domain observations for the atmosphere and ocean. However, since the atmosphere and ocean share the same outer loop, their updated analysis will be used to acquire the new model trajectory for the next outer loop. This can improve the atmosphere and ocean analyses through the dynamical coupling at the data assimilation step.

Based on the definition by Penny et al. [2017], which defines Quasi-Strongly Coupled DA as "observations are assimilated from a subset of components of the coupled system. The observations are permitted to influence other components during the analysis phase, but the coupled system is not necessarily treated as a single integrated system at all stages of the process", the outer loop coupling used by CERA belongs to the quasi-SCDA method.

We have revised L65 as

"The European Centre for Medium-Range Weather Forecasts (ECMWF) implemented the "outer loop coupling", where the incremental 4D-Var atmospheric and 3D-Var with the First Guess at the Appropriate Time (3D-FGAT, Lee et al., [2004]; Lawless [2010]) oceanic analyses share the same outer loops so that their updated analyses will be used together to acquire the new model trajectory for the next outer loop [Laloyaux et al., 2016; 2018]. **Though cross-domain observations are not directly assimilated into separate earth components, separate earth component analysis benefit from a more coherent coupled-state through the dynamical coupling at the data assimilation step. Based on Penny et al. [2017], outer loop coupling belongs to the *Quasi-SCDA* methods.**"

- **ln 222 - 224:** They talk about an experiment that uses 6-hour forcing in Figure 4 (a)-(b), but these subfigures do not present such experiment.

**R:** Thanks for the comments. For the 3D-Var UCDA experiment, the most frequent forcing update is 1-day (experiment UC_1day), not 6-hour forcing experiment. we choose the slowest forcing update as 1 day to mimic the common approach adopted by operational centers where their uncoupled atmosphere model uses the daily SST products as the surface forcing.

We have revised L245 as:

"However, the analysis error with a **1-day** forcing update is still one order of magnitude greater …"

- **Figure 7:** The labels of the subfigures are wrong. Panels (c)-(d) do not exist.
**R:** We have revised the labels of the subfigures in Figure 7.

- **ln 338:** It is mentioned the result for WCDA experiment, but the figure does not show it, or it is not clear.
**R:** We have revised the caption of Figure 9. Now the improvement/degradation by switching from WC to SC CFSv2-LETKF are shown in Figure 9.

Technical corrections:
- Make sure all the captions have the same style for the citations.
**R:** We have revised the citation styles in the captions of Figures 1, 8, 9, 10, 12.

- **ln 63**: Citation for the 3D-FGAT method is needed.
**R**: 3D-Var with the First-Guess at the Appropriate Time (3D-FGAT, Lee et al., 2004; Lawless, 2010)
We revise L66 "…, where the incremental 4D-Var atmospheric and 3D-FGAT oceanic analyses …"

as

"…, where the incremental 4D-Var atmospheric and 3D-Var with the First Guess at the Appropriate Time (3D-FGAT, Lee et al., [2004]; Lawless [2010]) oceanic analyses…"
Reference:
Lee, M.-S. D. Barker, W. Huang, and Y.-H. Kuo: First guess at appropriate time (FGAT) with WRF 3DVAR. *WRF/MM5 Users Workshop*, pp. 22-25. 2004.

Lawless A. S.: A note on the analysis error associated with 3D-FGAT. *Q. J. R. Meteorol. Soc.* **136**: 1094–1098, doi: 10.1002/qj.619. 2010.

- **ln 104:** "... Kalnay [2004], of which equations are written as." Replace the period with a colon.
**R:** Corrected.

- **ln 123:** "…standard deviation of √2. Besides, Assimilation experiments…" Change
capital to low-case in word 'Assimilation'.
**R:** Corrected.

- **ln 124:** "... experiments with the Ensemble Transform Kalman Filter (ETKF) in this section uses 9 members." Mismatch between subject and verb. Change "uses" to "use".

**R:** Corrected.

- **ln 177:** "..., and constant surface fluxes [$f_{X,i}$, $f_{Y,i}$, $f_{Z,i}$] that forces ..." Change "forces" to "force",

**R:** Corrected.

- **ln 178:** "... $x_0^b$ represents the initial background states, ..." Change "states" for "state".

**R:** Corrected.

- **ln 183:** NMC is an acronym not defined.

**R:** NMC method refers to "National Meteorological Center" method (Parrish and Derber, 1992).

We have revised L184: "Where $\mathbf{B}_{x,0}$ is the background error covariance of the initial ocean states estimated by the NMC methods"

to

"Where $\mathbf{B}_{x,0}$ is the background error covariance of the initial ocean states estimated by the National Meteorological Center (NMC) method [Parrish and Derber, 1992]."

Reference:
Parrish, D. F, and Derber J.C.: The National Meteorological Center's spectral statistical interpolation analysis system. *Mon. Weather Rev.,* 120:1747-1763

- **ln 225:** "... update is still one order greater than... " Change to "... update is still one order of magnitude greater than... "

**R:** Corrected.

- **ln 287:** Acronym OSSE is not defined.

**R:** We revise L310 "A 6-year perfect model OSSE is then conducted …"

To

"A 6-year perfect model **Observation System Simulation Experiment (OSSE)** is then conducted…"

- **ln 313:** missing indent at beginning of paragraph.

**R:** Thanks for the comment. Since this sentence is the 1st sentence under Section 6, we do not indent. To make the style consistent through the whole context, we also remove the indentation before the 1st sentence under Section 5.

---

## Author Response (AR2)

We thank the editor for his constructive suggestions!

I only have a few remarks about aspects that the authors might address in a better way in the manuscript, as listed below:
- Figure 1: the quality of the figure shall be improved, as recommended by reviewer 2. So far, the resolution in the revised draft is still quite low;

Thank you for this suggestion.
We have added a higher-resolution Figure 1.
Besides, we also replaced the original Figure 2 with a higher-resolution one.

- Figure 4 and discussion: in their reply to reviewer 1, the authors extend their description about comparing different methodologies to assimilate in the ocean. I would recommend that this discussion is also provided in the draft (especially regarding the table and figure in the authors' reply);

We have added the discussion about the table and figure (**in bold**) to the context:

"However, the analysis error with a 1-day forcing update is still one order of magnitude greater than the ocean analyses obtained from the coupled models. **For the last ~11 model years, the WC 3D-Var achieves an averaged analysis RMSE of $1.160\times10^{-3}$ for the atmosphere and $5.516\times10^{-5}$ for the ocean. For the SC 3D-Var, the corresponding analysis RMSE is $1.159\times10^{-3}$ for the atmosphere and $4.915\times10^{-5}$ for the ocean, both smaller than the error from the WC 3D-Var.** Among all three CDA configurations, SCDA analyses are the most accurate for the coupled states. **Besides, the SC 3D-Var shows lower RMSE than the WC 3D-Var for the ocean during the spin-up period, and the SC 3D-Var also experiences a shorter spin-up period (Figure not shown).**"